# Arbuscular-Mycorrhizal Symbiosis in *Medicago* Regulated by the Transcription Factor MtbHLHm1;1 and the Ammonium Facilitator Protein MtAMF1;3

**DOI:** 10.3390/ijms241814263

**Published:** 2023-09-19

**Authors:** Evgenia Ovchinnikova, David Chiasson, Zhengyu Wen, Yue Wu, Hero Tahaei, Penelope M. C. Smith, Francine Perrine-Walker, Brent N. Kaiser

**Affiliations:** 1School of Life and Environmental Sciences, The University of Sydney, 380 Werombi Road, Brownlow Hill, NSW 2570, Australia; 2Department of Biology, Saint Mary’s University, Halifax, NS B3H 3C3, Canada; 3School of Agriculture, Food and Wine, Waite Campus, University of Adelaide, Urrbrae, SA 5005, Australia; 4Agribio, Centre for AgriBiosciences, La Trobe University, 5 Ring Road, Bundoora, VIC 3083, Australia; 5Sydney Institute of Agriculture, The University of Sydney, 380 Werombi Road, Brownlow Hill, NSW 2570, Australia

**Keywords:** bHLHm1;1, AMF1, ammonium, phosphate, arbuscule, mycorrhiza, legume

## Abstract

Root systems of most land plants are colonised by arbuscular mycorrhiza fungi. The symbiosis supports nutrient acquisition strategies predominantly associated with plant access to inorganic phosphate. The nutrient acquisition is enhanced through an extensive network of external fungal hyphae that extends out into the soil, together with the development of fungal structures forming specialised interfaces with root cortical cells. Orthologs of the bHLHm1;1 transcription factor, previously described in soybean nodules (GmbHLHm1) and linked to the ammonium facilitator protein GmAMF1;3, have been identified in Medicago (*Medicago truncatula)* roots colonised by AM fungi. Expression studies indicate that transcripts of both genes are also present in arbuscular containing root cortical cells and that the MtbHLHm1;1 shows affinity to the promoter of *MtAMF1;3*. Both genes are induced by AM colonisation. Loss of *Mtbhlhm1;1* expression disrupts AM arbuscule abundance and the expression of the ammonium transporter *MtAMF1;3*. Disruption of *Mtamf1;3* expression reduces both AM colonisation and arbuscule development. The respective activities of MtbHLHm1;1 and MtAMF1;3 highlight the conservation of putative ammonium regulators supporting both the rhizobial and AM fungal symbiosis in legumes.

## 1. Introduction

Most land plants form arbuscular mycorrhiza through a symbiosis with soil fungi from the phylum Glomeromycota, collectively called arbuscular-mycorrhizal (AM) fungi. All AM fungi are obligate biotrophs as they rely on the plant host to survive and propagate [1,2]. Colonisation of roots occurs via hyphae, growing from spores or previously colonised roots, which branch extensively near plant roots in response to root exudates. On the root surface, hyphae form hyphopodia through which the AM fungus penetrates the root epidermis eventually colonising the root cortex [3]. In many AM symbioses, cortical cell penetration involves a process where the host plasma membrane invaginates and proliferates around the fungal intrusion. Repeated dichotomous branching of the intracellular fungal hyphal trunk produces symbiotic structures, called arbuscules, which provide complex interfaces between fungus and plant, across which nutrient exchange proceeds between the plant and fungal partners. Coils and intermediate hyphal structures can also develop often in a strain-dependent manner [2].

Knowledge of signaling pathways linked to AM establishment and development in plants is expanding and has been reviewed by Pimprikar and Gutjahr [4]. Apart from the collection of transcription factors (TFs) which are part of the ‘common symbiosis signaling’ network shared between symbioses formed by root nodule bacteria and AM fungi [4,5,6,7,8,9,10], there are a number of AM mycorrhizal TF networks, which show tentative linkages to the nutritional functionality of AM symbioses. Xue et al. [11] identified an *AP2 family TF*, the *CCTF MOTIF-BINDING TRANSCRIPTION FACTOR1 (CBX1)* in *Lotus japonicus,* which binds to conserved *CTTC* promoter motifs of AM fungal enhanced genes including *LjPT4, LjHA1* and *LjRAM2.* A similar *AP2/EREBP* transcription factor family (*WR15a, WR15b/Erf1,* and *WR15a*) was identified in *Medicago truncatula* and shown to bind to AW-box *cis*-regulatory elements in the promoter regions of the periarbuscular lipid transfer gene *MtSTR* and *MtPT4* [12]. *MtPT4* encodes a high-affinity Pi transporter expressed only when in symbiosis with AM fungi and only found localised to the periarbuscular membrane (PAM) in arbuscule-colonised cortical cells [13,14]. MtPT4 is believed to transfer Pi from the periarbuscular space across the PAM to the host cell. It belongs to a class of the Pi transporters which are specifically expressed in mycorrhizal roots and which retain the *CTTC CRE* motif in their promoters. *LjHA1* encodes an H^+^-ATPase [15] deemed essential for establishing proton gradients across the PAM to facilitate LjPT4-mediated Pi transport. *RAM2* encodes an endoplasmic reticulum localised glycerol 3-phosphate acyl transferase involved in the biosynthesis of plant lipids required for arbuscule development [16,17,18].

Transcriptional networks regulating ammonium transport remain poorly understood in plant roots and those colonised by AM fungi [19]. Xuan et al. [20] have described the interactions between *INDETERMINATE DOMAIN 10 (IDD10)* and the promoter of *OsAMT1;2*, a member of the high-affinity *(AMT/MEP/Rhesus)* transport family [21]. In this example, binding by IDD10 enhances the expression of *AMT1;2* and other N-linked genes. Recently, the DNA BINDING WITH ONE FINGER 18 *(DOF18*) TF has been linked to the expression of multiple *AMT* genes in rice [22]. Das et al. [23] and Shi et al. [24] in rice showed that phosphate starvation response (PHR) transcription factors regulated mycorrhizal symbiosis and symbiosis-related genes via the P1BS motif. By generating a transcriptional regulatory network for AM symbiosis in rice, Shi et al. [24] demonstrated that *OsPHR1/2/3* activated 25 out of the 51 target promoters which included a phosphate *(OsPT11)* and an ammonium transporter *(OsAMT3;1).* However, only *OsPHR2* displayed highly enriched expression in arbuscular-containing cells, and the overexpression of *OsPHR2* partially induced the expression of *OsAMT3* even in the absence of mycorrhizal infection [24]. Furthermore, Das et al. [23] identified three PHR transcription factor genes, *PHR1A*, *PHR1B*, and *PHR1C* in *L. japonicus* and the mutation of *Ljphr1a* caused a significant reduction in colonisation by *R. irregularis* as compared to the wild type of hairy roots transformed with an empty vector and when grown at low phosphate concentration (at 25 µM P_i_). In addition, the expression of *SYMRK*, *CCaMK* and *CYCLOPS* was reduced in roots of *phr1a* [23].

Other extensive studies on arbuscular mycorrhizal symbioses during the last few decades have highlighted its significance for both natural and agricultural ecosystems in the improvement of phosphorus acquisition under limiting soil environments [25]. Arbuscular mycorrhizas are often associated with symbiotic inorganic phosphorus (Pi) uptake and release into the plant [13,14], resulting in increased P uptake into AM plants compared with non-colonised controls. Work by Das et al. [23] and Shi et al. [24] suggest that there is a link between phosphate and ammonium acquisition during the AM symbiosis in rice and *L. japonicus*. Recently, evidence suggests that ammonium may also be transferred to plants via the mycorrhizal pathway [26,27,28,29,30], but the quantitative significance in overall plant nitrogen uptake remains unclear (see Smith and Smith [31] for discussion on limited evidence for net nitrogen delivery via AM systems). Prior work by Kobae et al. [32] identified 16 ammonium transporter (*AMT*) genes in the *Glycine max* genome and five of these *AMTs*, *GmAMT1.4*, *GmAMT3.1*, *GmAMT4.1*, *GmAMT4.3* and *GmAMT4.4* respectively were AM inducible. By hairy root transformation, soybean expressing *GmAMT4.1 promoter-GUS (β-glucuronidase)* and *GmAMT4.1 promoter: Gm4.1-GFP* (green fluorescent protein) constructs were inoculated with *Glomus mossae* and observed under light and confocal microscopy. A strong GUS activity was only detected in arbusculated inner cortical cells and it was confirmed by GmAMT4.1-GFP signal only in arbusculated cells [32]. The authors concluded that active ammonium transfer occurred around arbuscule branches due to the specific localisation of GmAMT4.1-GFP on the branch domain of the periarbuscucular membrane but not on the trunk region. Koegel et al. [33] found that the expression of two *AMTs* in *Sorghum bicolor*, *SbAMT3;1* and *SbAMT4*, were induced in root cells containing arbuscules and in adjacent cells and *SbAMT3;1* was present only in cells containing developing arbuscules by immunolocalisation studies. However, the role of this transport process remains poorly understood. The evidence for a functional transport link between the AM symbiosis and AMT2 transporters at the periarbuscular membrane is still unclear. Breuillin-Sessoms et al. [34] showed that N starvation is an effective suppressor of premature arbuscule degradation (PAD) commonly observed in *Mt-pt4* mutants. AMT2.3 activity was required for PAD suppression but loss of *amt2.3* had no influence on shoot N content when in symbiosis. Recently, experimental studies by Wang et al. [35] in *L. japonicus* observed that *LjAMT2;2* expression was upregulated by *Glomus intraradices* at 28 days of symbiosis and its expression was also upregulated in roots with symbiotic mycorrhiza in a low-nitrogen environment. Furthermore, at a high ammonium ion concentration (4 mM), *LjAMT2;2* expression decreased in roots inoculated with *G. intraradices* suggesting that the decrease in gene expression could be a mechanism to avoid ammonium toxicity in a high-nitrogen environment [35].

In soybean, Chiasson et al. [36] characterised a membrane-bound transcription factor GmbHLHm1 (SAT1), as an important component of the nitrogen-fixing symbiosis. GmbHLHm1 was identified by its ability to transcriptionally control an endogenous yeast ammonium transport facilitator *ScAMF1* (YOR378W) [36,37]. *ScAMF1* is the homolog of the DHA2 family of the drug: H^+^ antiporters involved in a range of transport processes including, ammonium, boron, siderophores, aminotriazole, and glutathione [38,39]. A soybean *AMF1* orthologue, *GmAMF1;3* was also shown to be expressed in root nodules and capable of low-affinity ammonium transport. Although symbiotically enhanced, these proteins are present in all higher plants including those that do not develop a rhizobial symbiosis. Given the commonalities revealed between signaling pathways in arbuscular mycorrhizal and rhizobial symbioses [40,41], we investigated whether orthologues of these two genes are involved in AM symbioses.

Here, we focused on the model legume *Medicago truncatula* using contemporary molecular genetics to demonstrate if *MtbHLHm1;1* and *MtAMF1;3* are involved in the *Medicago* AM fungal symbiosis at either a signaling or developmental context. Plant symbiotic development, gene expression, reverse genetics, and intercellular protein targeting has been used to test the responsiveness of *MtbHLHm1;1* and *MtAMF1;3* to AM fungal colonisation, inorganic Pi supply, and activity. This study has highlighted that both *MtbHLHm1;1* and *MtAMF1;3* are involved in the AM symbiosis in *Medicago*, respond in a Pi-sensitive manner, and shed new insights into potential ammonium-dependent processes operating in AM fungal colonised roots.

## 2. Results

### 2.1. Both bHLHm1 and AMF Classes of Proteins Are Found in Mycorrhizal and Non-Inoculated Plants

An amino acid homology search in Medicago identified MtbHLHm1;1 (Medtr2g010450) as a close orthologue (82.4% identity) of GmbHLHm1 (Glyma15g06680) and MtAMF1;3 (Medtr2g010370.1) as a close orthologue (75% identity) of GmAMF1;3 (Glyma08g06880). Both genes are found in higher plants and share up to 90% amino acid sequence identity amongst other legumes and up to 72% identity with other plants including non-mycorrhizal species such as *Arabidopsis thaliana* (Appendix A). We found that MtbHLHm1;1 is part of a legume-specific clade that includes GmbHLHm1 (Appendix A). MtAMF1;3 is also clustered into a legume-based clade that contains GmAMF1;3 (Appendix A). In *Medicago*, we have identified two bHLHm1;1 members (MtbHLHm1;1 and MtbHLHm1;2), three AMF1 members (MtAMF1;1–MtAMF1;3), and a single member of a predicted AMF2 clade (MtAMF1;2) (Appendix A).

### 2.2. MtbHLHm1;1 and MtAMF1;3 Are Differentially Expressed in Mycorrhizal and Unincoulated Roots

To investigate the relationship between *MtbHLHm1;1*, *MtAMF1;3* and mycorrhiza in *Medicago*, we tested for changes in gene expression when transgenic plants were supplied low Pi (50 µM) and inoculated with a combined arbuscular-mycorrhizal (AM) fungal spore population (*Glomus etunicatum*, *Glomus coronatum*, *Glomus intraradices* and *Glomus mosseae*). Transgenic (hairy-root) plant roots were evaluated 8 weeks after inoculation (WAI). In empty vector (control) roots, *MtbHLHm1;1*, *MtAMF1;3,* and *MtAMF2;1* expression were significantly (*p* < 0.05) enhanced when inoculated with AM fungi (Figure 1A). We also tested the expression of ammonium transporters belonging to the *AMT1* and *AMT2* families [42,43,44,45]. *MtAMT1;3* and *MtAMT2;4* were significantly upregulated in AM roots compared to non-AM control roots (Figure 1A). As a control, we also examined the expression pattern of the AM-induced inorganic Pi transporter, *MtPT4* in both non and AM colonised Medicago roots (Figure 1A). As shown previously by Javot et al. [46], colonisation by AM fungi enhanced *MtPT4* expression in roots confirming the symbiotic pathway was active and that AM colonisation was occurring.

We then silenced *MtbHLHm1;1* using the hairy-root technique. Wild-type *Medicago* plants (A17) [47] were transformed with an empty RNAi vector (Control) or with a *35S_pro_::Mtbhlhm1;1* construct (RNAi) using a hairy-root transformation system we established previously in soybean [48]. Total RNA was extracted from (control, ±AM) and (RNAi, ±AM) harvested roots of each event and analysed by qPCR for fold-changes in *MtbHLHm1;1* gene expression. RNAi silencing significantly decreased the expression of *MtbHLHm1;1* in the AM colonised roots (Appendix A). *35S_pro_::Mtbhlhm1;1* reduced expression of *MtbHLHm1;1* from ~9.9-fold in the control roots to less than ~1-fold in the RNAi roots (*p* < 0.007) (Figure 1A,B, Appendix A). With the loss of *Mtbhlhm1;1*, the expression of *MtAMF1;3* was no longer upregulated after AM colonisation (Figure 1B). The specificity of *35S_pro_::Mtbhlhm1;1* was also tested against other genes including its close homolog *MtbHLHm1;2* and a *MtAMF1;3* homolog, *MtAMF1;1* (Appendix A). *35S_pro_::Mtbhlhm1;1* also silenced *MtbHLHm1;2* and *MtAMF1;1*. A control construct, *35S_pro_::Mtbhlhm2*, was also developed and tested for its impact on *MtbHLHm1;1* and *MtAMF1;3* expression. *Mtbhlhm2* reduced the scale of the +AM response for *MtbHLHm1;1*, *MtbHLHm2*, and *MtAMF1;1* expression, while eliminating all +AM responsiveness in *MtAMF1;3* (Appendix A). We can conclude that *MtbHLHm1;1* and *MtbHLHm1;2* have some redundancy and both influence the expression of *MtAMF1;3*.

The reduction in *Mtbhlhm1;1* expression enhanced the AM-induced expression of *MtAMT1;2*, *MtAMT2;1*, *MtAMT2;2* and *MtAMT2;4* relative to the -AM controls (Figure 1B, *p* < 0.05). The loss of AM enhanced expression of *MtAMF1;3*, *MtAMF2;1* and the ammonium transporter *MtAMT2;3* [34] in *Mtbhlhm1;1* RNAi silenced roots suggests a potential role of MtbHLHm1;1 in the molecular control of multiple ammonium transport pathways in mycorrhizal colonised roots. Loss of *Mtbhlhm1;1*, reduced the AM-induced expression of *MtPT4* (Figure 1B). This result highlights the significant impact of MtbHLHm1;1 on the AM symbiosis and the settings that support AM-induced expression of *MtPT4* in arbuscules containing root cortical cells.

### 2.3. MtbHLHm1;1 Affects Mycorrhizal Arbuscule Abundance and AM Pi Responses in Roots

We then investigated the relationship between MtbHLHm1;1, mycorrhiza colonisation, and exogenous Pi supply. Transgenic (Control or *Mtblhlm1;1*) roots were inoculated with a combined arbuscular-mycorrhizal (AM) fungal spore population and then grown for a further 6 weeks on low Pi (50 µM). At 6 weeks, the plants were left at either 50 µM Pi or moved to a higher Pi concentration (500 µM) for 2 weeks to help repress the mycorrhizal symbiosis [50,51]. We found *MtbHLHm1;1* expression was significantly enhanced (~6 fold, *p* < 0.05) in non-inoculated AM roots exposed to high Pi (500 µM) relative to those grown at low Pi (50 µM) (Figure 2A).

Plant and shoot dry weights increased with AM inoculation in the *Mthlhm1;1* RNAi line; though root growth remained similar to the non-inoculated plants (Figure 3A–C). Collectively, the mycorrhizal growth response (%MGR) was significantly lower in the *Mtbhlhm1;1* RNAi line (Figure 3D). The AM fungal structures in the *Mtbhlhm1;1* RNAi line were visualised using the ink staining method according to [52] and scored (Figure 2B–D) using the grid intersect method as described in [53]. Roots were evaluated for total mycorrhizal colonisation (%MC, which includes hyphae, arbuscules, and vesicles) and arbuscule abundance (%Arb) (Figure 2B,C). In general, control plants showed a significant reduction in %MC and %Arb when supplied 500 µM Pi (Figure 2B,C). The loss of *Mtbhlhm1;1* significantly decreased %MC (*p* = 0.0006), though the impact was similar at low and high Pi treatments (Figure 2B). We found no significant change in %Arb between the control and *Mtbhlhm1;1* at either Pi concentration (Figure 2C). %Arbuscule degradation increased in both the empty vector control and the *Mtbhlhm1;1* roots in response to elevated Pi (500 µM) (Figure 2D).

We then developed knockdown lines for MtAMF1;3 (*35S_pro_::Mtamf1;3*) using RNAi (Appendix A). In the presence of AM fungi and 50 µM Pi, empty vector (control) transformed plants responded to AM colonisation through an increased plant, shoot, and root dry weights (Figure 3A–C). The AM enhanced growth was reduced with RNAi silencing of *Mtamf1;3* (*35S_pro_::Mtamf1;3*) lines when presented as either total plant dry weight, shoot dry weight, and root dry weight (Figure 3A–C). %MGR, %MC and %Arb all decreased significantly to the control lines (Figure 3D–F, respectively). Ink-stained overviews of AM colonised roots of *Mtamf1;3* show a general reduction in colonisation and arbuscule presentation relative to empty vector controls (Appendix A).

### 2.4. Cellular Localisation of MtbHLHm1;1 and MtAMF1;3 in AM Fungal Colonised Roots

To localise the cellular expression of *MtbHLHm1;1* and *MtAMF1;3* in AM roots, we used promoter GUS (β-glucuronidase) reporters. In parallel, we also used constitutively expressed N-terminal fusions with green fluorescence protein (GFP) to identify the default intercellular localisation of overexpressed *MtbHLHm1;1* and *MtAMF1;3* proteins. Hairy roots of Medicago plants were generated with constructs containing *MtbHLHm1;1_pro_:GUS*, *MtAMF1;3_pro_:GUS*, *UBQ3_pro_:GFP-MtbHLHm1;1* and *UBQ3_pro_:GFP-MtAMF1;3*. Transgenic roots were then inoculated with AM fungi and analysed at 8 weeks after inoculation (WAI). Non-inoculated plants of similar age were used as a control. *MtbHLHm1;1pro* and *MtAMF1;3pro* were found active in both non-inoculated and mycorrhizal roots (Figure 4A,B and Figure 5A,B, respectively). In the non-inoculated control roots, GUS expression for both *MtbHLHm1;1* and *MtAMF1;3* were classified as diffuse across the epidermis, root hair, cortex, and stele (Figure 4A and Figure 5A, respectively). In mycorrhizal colonised roots, GUS staining was more specific and localised in cortical cells and the stele (Figure 4B), where AM fungi were also identified using secondary ink-staining (Figure 4C) or with WGA-Texas Red antibody (Figure 5D,E). Quantitative expression levels of GUS were not measured in these plants, instead, we used more sensitive mRNA transcript abundance (qPCR) assays to record the positive influences on *MtbHLHm1* and *MtAMF1;3* gene expression by AM colonisation (Figure 1).

The *UBIQUITIN 3* promoter [54] was then used to express *GFP-MtbHLHm1;1* and *GFP-MtAMF1;3* mRNA in *Medicago* roots to help identify the default intercellular targeting of both encoded proteins (Figure 4D–I and Figure 5F–M, respectively). In non-inoculated roots (control), GFP-MtbHLHm1;1 was found targeted to the nuclei (Figure 4D,E) while GFP-MtAMF1;3 protein was present in cortical and epidermal cells including root hairs (Figure 5F,G,J). In mycorrhizal roots, GFP-bHLHm1;1 extended to nuclei and to putative membranes of cortical cells containing arbuscules (Figure 4F–I). GFP-MtAMF1;3 protein was found aligned to membrane locations of cortical, stele, and epidermal cells and associated with membranes of cortical cells containing arbuscules (Figure 5H,I,K–M). The lack of protein-specific antibodies for this study has limited our ability to conclusively identify the intercellular membrane locations of both MtbHLHm1 and MtAMF1;3.

### 2.5. MtbHLHm1;1 and MtAMF1;3 Influence Ammonium Transport in Yeast Cells

We tested if MtbHLHm1;1 and MTAMF1;3 share ammonium transport activities in common with their soybean orthologues when expressed in the *Saccharomyces cerevisiae* ammonium transport mutant 26972c (*mep1 mep2Δ Mep3*). This mutant lacks an active methylammonium/ammonium permease (MEP) transport system [55] as a result of a deletion of *MEP2* and a mutation in *MEP1* that disrupts MEP3 activity. These mutations limit the ability of 26972c to accumulate methylammonium (MA) and develop associated toxicities when yeast cells are exposed to high concentrations (>50 mM MA). We previously demonstrated that GmbHLHm1;1 can bind the *ScAMF1* promoter, a process required to complement the 26972c yeast strain and facilitate increased MA uptake into the cell [36]. *ScAMF1*, *MtbHLHm1;1* and *MtAMF1;3* were each introduced into 26972c under control of the galactose-inducible (GAL1) promoter. In each case, 26972c cells exposed to 0.1 M MA showed reduced growth relative to cells containing the empty vector (pYES3) on a plate (Figure 6A) and liquid culture (Figure 6B vs. Figure 6C). Reduced growth in the presence of MA was accompanied by a significant increase in uptake of ^14^C-MA into 26972c cells expressing either *MtbHLHm1;1* (1.7-fold, *p* < 0.05) or *MtAMF1;3* (2-fold, *p* < 0.05) relative to the empty vector control *pYES3* (Figure 6D). The rate of uptake was similar to the positive control, ScAMF1. These results indicate that both MtbHLHm1;1 and MtAMF1;3 enhance MA uptake in the 26972c mutant and that MtAMF1;3 specifically transports MA when expressed in yeast cells.

We tested if MtbHLHm1;1 recognised the promoter of *MtAMF1;3* using purified bHLHm1;1 protein in an electromobility shift analysis experiment (Figure 6E). When added to a DIG-labelled promoter sequence of *MtAMF1;1*; which contains conserved *E-box* DNA binding domains, we observed selective binding of the promoter by MtbHLHm1;1 and retardation of the gel fragment (Figure 6E). This result was similar to that observed by Chiasson et al. [34], where GmbHLHm1;1 was able to bind to the *E-box* containing promoter of *ScAMF1*. This result suggests that MtbHLhm1;1 is a probable DNA binding transcription factor capable of recognising the *MtAMF1;3* promoters.

## 3. Discussion

In this study, we have characterised the involvement of the bHLHm1 transcription factor MtbHLHm1;1 and an AMF1 ammonium facilitator homolog, MtAMF1;3 in the development and activity of AM fungal colonised *Medicago* roots. The results indicate both proteins are required for the development of the arbuscular mycorrhizal symbiosis in *Medicago* to varying degrees. The TF, *MtbHLHm1;1,* and members of the *MtAMF* transport family were found upregulated in mycorrhizal colonised roots. Overexpression of *MtbHLHm1;1* or *MtAMF1;3* in the yeast ammonium transport mutant (26972c) revealed a similar phenotype to that observed previously for GmbHLHm1;1, ScAMF1 and GmAMF1;3: sensitivity to high concentrations of MA, ability to accumulate a net quantity of ^14^C-MA and an affinity of the TF MtbHLHm1;1 protein to the promoter of *MtAMF1;3*. [36]. In AM colonised *Mtbhlhm1;1*- RNAi silenced roots, both *MtAMF1* and *MtAMT2* transporters were found down-regulated. The *MtbHLHm1;1* promoter also delivered a GUS signal to +AM colonised root cortical cells containing arbuscules. Using GFP-tagged MtbHLHm1;1, we show that overexpressed protein is targeted by default to the nucleus in ±AM roots but also to arbuscule structures located inside root cortical cells. GmbHLHm1;1 also has a multiple targeting capacity that includes the symbiosome and plasma membranes but also the ER and nucleus [36,37]. Knock-down of *Mtbhlhm1;1* or *Mtamf1;3* resulted in reduced mycorrhizal colonisation. Overexpressed GFP-tagged MtAMF1;3 protein was also located with MtbHLHm1;1 in arbuscule containing cortical cells, while electromobility shift analysis revealed the potential for direct transcriptional interaction between MtbHLHm1;1 and the promoter of *MtAMF1;3*. Together, these data suggest a role for both MtbHLHm1;1 and MtAMF1;3 in regulating mycorrhizal colonisation and linking a potential interaction between bHLHm1 and AMF1;3 activity in plant AM mycorrhizal fungal symbioses.

### 3.1. MtbHLhm1;1 and MtAMF1;3 Are Required for the AM Fungal Symbiosis in Medicago

*MtbHLHm1;1* is a close orthologue of the membrane-bound bHLHm1 transcription factor *GmbHLHm1*, previously characterised in soybean nodules and roots [36,37]. GmbHLHm1 was first identified by its ability to transcriptionally upregulate an uncharacterised plasma membrane localised low-affinity ammonium transport protein, *ScAMF1* (YOR378w) which assisted in the rescue of a yeast ammonium transport mutant (26972c) as well as inhibiting its growth on high levels of the toxic ammonium analogue, methylammonium [36,37]. ScAMF1 and a soybean homolog GmAMF3 were shown to act as ammonium transporters, facilitating both ammonium and methylammonium transport when expressed in either yeast cells or *Xenopus laevis* oocytes. Subsequent analysis of plant genomes identified multiple *AMF1* homologs in other plant species including two genes in *Medicago* (Appendix A). AMF proteins are distant homologs of the H^+^/Drug antiporters belonging to the DHA2 subfamily [56,57,58]. In yeast, multiple DHA2-like proteins have been directly or indirectly implicated in intracellular NH_4_^+^ homeostasis [38].

Across many dicot plant species, synteny at the chromosomal level for *bHLHm1* and *AMF1* homologs exists, where genes are found to be co-localised at common loci [36]. Here a detailed analysis of legume genomic sequences allowed us to identify two *Medicago* homologs with strong sequence homology to *GmbHLHm1* and *GmAMF1;3*. We confirmed both of these genes were expressed at basal levels in non-mycorrhizal roots and upregulated when colonised by AM fungi (Figure 1A). Using GUS, this change in expression was demonstrated by a refined cellular localisation of promoter-driven GUS expression (Arbuscules and the root stelle). This response was associated with several other genes examined including different *MtAMF* homologs, genes belonging to the high-affinity ammonium transporter family (*AMT*/*MEP*/*Rhesus*) *MtAMT1* and *MtAMT2* and the periarbuscular membrane (PAM) localised inorganic Pi transporter gene, *MtPT4* (Figure 1A). Promoter GUS fusions and UBQ=-driven GFP protein tagging approaches showed that both *MtbHLHm1;1* and *MtAMF1;3* were positioned in arbuscule containing root cortical cells; however, promoter activity or the internal cellular partitioning of protein was not exclusive across root cells (Figure 4F–I and Figure 5K–M). In addition, observations of GFP-tagged MtAMF1;3 protein localisation in arbuscular cortical cells were similar to observations made in other AMT studies by Breuillin-Sessoms et al. [34], Guether et al. [28], Kobae et al. [32], and Koegel et al. [33]. However, there are differences, such as the localisation of GFP-tagged MtAMF1;3 proteins in regions of the cell near the plasma membrane of root hairs and cortical cells of non-mycorrhizal roots, an outcome most likely linked to the MtAMF1;3 proteins not being under the control of its own promoter. Work by Guether et al. [28] also demonstrated using RT-PCR analysis, that *LjAMT2;2* was expressed in not only arbusculated cells but also in non-colonised cortical cells from mycorrhizal roots and cortical cells from nonmycorrhizal roots. This suggested that there is a possible role for MtAMF1;3 in root cell types other than arbusculated cortical cells in *Medicago*. Gaude et al. [59] highlighted that during arbuscular mycorrhizal development, *Medicago* roots underwent extensive and specific reprogramming and, transcripts encoding proteins such as bHLH, GRAS or Myb family transcription factors were up-regulated in arbuscule-containing cells compared to non-arbuscule containing cells. We suggest similar activity is occurring with MtbHLHm1;1 in response to AM colonisation.

The promoter GUS fusions indicated a concentration in cortical cellular expression with AM colonised cells for both genes (Figure 4B). The RNAi-mediated knockdown of *Mtbhlhm1;1* caused a reduction in %Arb but the differences between low and high Pi treatments were similar relative to the control roots (Figure 2C). Interestingly, the loss of *Mtbhlhm1;1* overcame the Pi-enhanced reduction in %MC seen in control roots (Figure 2B). The loss of *Mtbhlhm1;1* expression in roots was also associated with a significant reduction in the expression of *MtAMF1;*3 and a corresponding derepression of *MtAMT2;1* and *MtAMT2;2* (Figure 1B). There was no change in expression of either *MtAMT2;3* or *MtAMT2;4* (relative to −AM controls), two genes that have been linked with the maintenance of arbuscules in *Medicago* colonised roots in a manner linked to the nitrogen status of the symbiosis [34]. The loss of *MtAMF1;3* expression consistently disrupted growth and AM colonisation (plant DW, %MGR, %MYC and %Arb).

Observations in *Medicago* and *L. japonicus* with mutated bHLH transcription factors have been shown to affect nodulation by impairing root hair development [60], causing vascular defects in nitrogen-fixing nodules [7], and negatively regulating nodule senescence [5]. In the latter, a MtbHLH2 was shown to be a transcriptional repressor localised in the nucleus and by transcriptome data analysis identified a papain-like Cys protease gene, *MtCp77*, as its potential target [5]. Further, independent analysis by chromatin immunoprecipitation demonstrated that a separate MtbHLH2 bound directly to the promoter of *MtCP77* to inhibit its expression [5]. MtCP77 was shown to positively regulate nodule senescence by accelerating plant programmed cell death (PCD) and accumulation of reactive oxygen species (ROS) and *bhlh2* nodulated plants displayed nodules with early nodule senescence and decreased nitrogen fixation [5]. Thus, in our case, *Mtbhlhm1;1* roots would no longer have a functional MtbHLHm1;1 TF to influence the expression of linked genes, including *AMF*, and *AMT* in the AM-inoculated roots. This loss of MtAMF1;3 activity appears to be the main driver of the phenotypes being observed in response to AM colonisation.

The interaction between *bHLHm1* and *AMF* genes have been previously observed in other systems [36]. In yeast, GmbHLHm1 transcriptionally activated *ScAMF1* and *ScMEP3*, which together helped rescue the ammonium transport deficiency phenotype of the common yeast mutants 26972c or 31019b. The relationship between *GmbHLHm1* and *GmAMF1;3* is still poorly defined but both genes are co-expressed in similar cell types in nitrogen-fixing soybean nodules and both genes occupy syntenic loci across most dicot plant species. The loss of *GmbHLHm1* expression resulted in a reduction in *GmAMF1;3* expression in soybean nodules [36]. As noted above, both *MtbHLHm1;1* and *MtAMF1;3* are expressed in similar tissues and cell types while the expression of *MtAMF1;3* is linked to the activity of MtbHLHm1;1 (Figure 1). We suggest this interaction is controlled by a *trans* interaction between the MtbHLHm1;1 TF and the promoter region of *MtAMF1;3* as suggested using in vitro affinity assays and electromobility shift experiments (Figure 6). We have shown that a purified fusion protein of MtbHLHm1;1 was able to bind to a DIG-labelled promoter region of *MtAMF1;3* at locations associated with *bHLHm1* e-box DNA binding domains (Figure 6E). This is a similar response to that observed with the direct binding of GmbHLHm1 to the promoter of *ScAMF1* in yeast supporting a conserved affinity of the bHLHm1 TF to the promoters of *AMF* encoding genes. In *Medicago*, we found that *MtAMF1;3* and a few *MtAMT2* genes were linked to the AM coordinated expression of *MtbHLHm1;1*. The link between MtbHLHm1;1 and the expression of *MtAMT2* genes has not previously been observed. However, distantly related ammonium transporter genes *ScMEP1* and *ScMEP3* (both *AMT2* homologs) in yeast were upregulated following the expression of *GmbHLHm1* [36] in yeast cells.

*MtbHLHm1;1* expression is strongly induced by Pi supply in non-inoculated roots (Figure 2A). High expression was linked to a reduction in %MYC and %Arbuscules in control roots (Figure 2B,C). Elevated Pi increased the % of degraded arbuscules. However, the loss of *Mtbhlhm1;1* reversed the drop in %MC (Figure 2B) but had no impact on %Arb in AM colonised roots at either 50 or 500 µM Pi (Figure 2C). We saw no change in the abundance of degraded arbuscule structures with 500 µM Pi (Figure 2D). These results with the controls are like that observed by many other AM studies where %MC and %Arb are often reduced by increased exposure to high Pi concentrations [25,51,61,62]. The Pi-independent changes in %MC and %Arb with the loss of *Mtbhlhm1;1* suggests a direct relationship between the MtbHLHm1;1 TF and AM development. Zeng et al. [63] found that Pi- deficiency altered the expression of many TFs and Pi-responsive TFs belonging to bHLH or nuclear factor Y (NF-Y) families, showing induction under Pi-sufficient conditions (500 µM KH_2_PO_4_) or repression with Pi-deficient conditions (0 mM KH_2_PO_4_) in soybean. It is well known that high Pi concentrations negatively affect the development of arbuscular mycorrhizal symbiosis in plants [51,64,65]. However, recent work by Zhang et al. [66] has shown that an HLH domain-containing transcription factor, *RiPho4* from *R. irregularis,* was inhibited in its expression in mycorrhizal roots of *Eucalyptus grandis* supplied high Pi concentrations (300 and 1000 µM). When expressed in EY57 yeast cells, the GFP-RiPho4 fusion protein localised to the nuclei during low Pi treatment (600 µM and 1 mM K_2_HPO_4_). During high Pi treatment (10 mM K_2_HPO_4_), the fusion protein was found in the cytoplasm of EY57 yeast cells. Furthermore, total mycorrhizal colonisation in most virus-induced gene silencing (VIGS) *RiPho4-RNAi* tobacco roots was not significantly lower compared to control roots but there was a decrease in the mycorrhizal intensity and arbuscule abundance in *RiPho4-RNAi* roots where the arbuscules appeared abnormal or degenerating under 30, 100, and 300 µM Pi concentrations respectively [66]. Such a mechanism may be playing a role in our experiments on the mycorrhizae population at 500 µM Pi.

In a symbiotic context, only bHLHm1 has been linked to the transport of ammonium through its binding to the promoter of *ScAMF1* in yeast and the requirement of this binding to enact ammonium transport activities. We have confirmed here that similar to the interaction between GmbHLHm1 and *ScAMF1* in yeast, purified MtbHLHm1;1 can also bind to the promoter of *MtAMF1;3* (Figure 6E). The functional properties of MtbHLHm1;1 appear like those previously observed with *GmbHLHm1* when expressed in the ammonium/methylammonium transport deficient yeast mutant, 26972c. MtbHLHm1;1 enhanced the mutant strain’s susceptibility to toxic concentrations of MA (plate and liquid growth) and resulted in the accumulation of ^14^C-methylammonium ~1.7 times higher than in the empty vector control (*p* < 0.05, Figure 1). We expect that this trait is linked to the transcriptional upregulation of *ScAMF1* which is strongly upregulated by GmbHLHm1 [36]. Expression of *MtAMF1;3* also made 26972c cells more susceptible to methylammonium most likely due to its increased uptake, ~2.0 times higher than the empty vector control (*p* < 0.05) (Figure 1). Together with the previous interactions between, GmAMF1 and ScAMF1, MtAMF1 appears to facilitate the transport and accumulation of methylammonium into yeast cells and we suggest it most likely has a role in ammonium transport in *Medicago*. Unfortunately, we have no evidence to suggest a role in ammonium transport in AM colonised roots as that was outside the scope of this project. Previous work by Tanaka and Yano [67] demonstrated that N transfer to maize by the AM fungus, *Glomus aggregatum*, was dependent on the N form. By studying soil-ammonium-derived ^15^N to *S. bicolor* plants via *G. mosseae* mycelium in compartmented microcosms, Koegel et al. [68] demonstrated that there was a rapid ^15^N transfer between the symbiotic partners. Later studies by Koegel et al. [69] on the symbiotic interaction between *S. bicolor* and *G. intraradices* highlighted that both plant and fungal AMT gene expressions were affected by different forms of N and there was an interdependence of both partners in the mycorrhizal N uptake pathway. Future experiments could involve treating control and *Mtbhlhm1;1* mutant plants with different concentrations of ammonium ions in the presence of AM fungi and monitoring gene expression and nitrogen acquisition in the roots and shoots. This could determine if MtbHLHm1;1 and MtAMF1;3 play a role in preventing ammonium toxicity in plants at a cellular or tissue level.

In summary, we highlight the MtbHLHm1;1 TF in the regulation of the AM symbiosis in *Medicago truncatula*. MtbHLHm1;1 is a Pi-enhanced TF that in response to AM inoculation increases the expression of several genes linked to ammonium transport, including *AMT* and *AMF* genes. *MtbHLHm1;1* expression is linked to the activity of the low-affinity AMF transporter family, previously observed in soybean root nodules. Loss of *Mtamf1;3*, compromises AM development in *Medicago* roots. Cellular expression and protein localisation studies identify both *MtbHLHm1;1 and MtAMF1;3* are associated with root cells containing arbuscules. We suggest AMF activity influences ammonium transport in AM-colonised cells to support the symbiotic partnership (Appendix A).

### 3.2. Loss of MtAMF1;3 Activity Disrupts Mycorrhizal Activity

The presence of MtAMF1;3 appears important for the establishment and maintenance of AM fungal colonisation in *Medicago*. Knockdown experiments to reduce *MtAMF1;3* expression caused a drastic reduction in %MC (~50%, *p* < 0.05) and %arbuscule abundance (by ~75%, *p* < 0.05) in *Medicago* hairy roots (Figure 3). This suggests that AMF has a role in supporting the mycorrhizal symbiosis. This relationship is in line with other ammonium transport pathways recently characterised as part of the plant mycorrhizal symbiosis. Mutants of the *Medicago* symbiotic phosphate transporters *Mtpt4* and *Mtpt8* showed that ammonium transporters of the *MtAMT2* family recovered mutant mycorrhizal phenotypes, concluding that the symbiosis can be established without expression of the symbiotic Pi transporter pathway when sufficient ammonium transport activity occurs potentially across the PAM in arbuscule-containing cells [34]. The localisation of *MtAMF1;3* promoter-driven GUS activity and the intercellular targeting of UBQ-driven GFP-tagged MtAMF1;3 in arbuscular containing cells combined with an induction of expression when in symbiosis with AM fungi would suggest a similar role for MtAMF1;3 to members of the MtAMT2 family in the AM fungal plant symbiosis. Whether MtAMF1;3, works independently or is regulated alongside a number of MtAMT2 transporters still needs to be defined. What role either MtAMT2 or MtAMF1 have in the context of ammonium management of arbuscule-containing cortical cells still needs to be defined. As discussed by Smith and Smith [31], the role of the AM pathway in meeting plant nitrogen demands needs to be considered in light of the direct pathways operating via the roots. However, the management of potentially toxic levels of ammonium inside symbiotic cells does need to be considered for the longevity and stability of the symbiosis.

## 4. Materials and Methods

### 4.1. Plant Materials, AM Fungi and Growth Conditions

*Medicago truncatula* Gaertn. cv. Jemalong line A17 [47] was used in all experiments. A17 seeds were surface sterilised in 4% (*v/v*) sodium hypochlorite for 10 min and germinated in darkness on Fahraeus medium [70] agar plates: 24 h at 4 °C and 24 h at room temperature. After germination, seedlings were grown in growth chambers under a 16h/8h light/dark photoperiod at a 21 °C/18 °C light/dark temperature regime.

For all transgenic plant experiments, 5 day-old seedlings were subjected to *Agrobacterium rhizogenes* mediated transformation [48] using the K599 strain [71] as described in Limpens et al. [72], except that Fahraeus agar plates without antibiotics, were used in all procedures. Plants with transgenic roots were selected based on DsRED1 and/or GFP expression [72] using a Leica MZ10F modular stereo microscope fitted with the FLUOIII filter system (Leica). All non-transgenic roots were removed from each plantlet prior to AM inoculation.

A commercial inoculum of AM fungi (Start Up Super, MicrobeSmart Pty Ltd., Melrose Park, Australia) containing spores of *Glomus etunicatum*, *Glomus coronatum*, *Glomus intraradices,* and *Glomus mosseae* was used in all experiments to develop arbuscular mycorrhiza colonisation. Plants with transgenic “hairy roots” were inoculated with three grams of inoculum per pot (~3000 infective propagules g^−1^) and grown in a 1:1 perlite-vermiculite mixture, one plant per 0.5 L pot (10 cm high). Plants were watered twice a week and saturated with rhizobia-free sterile liquid nutrient solution: 0.0025 M ammonium nitrate, 0.0025 M microelements as in [70], 0.001 M iron citrate, 0.0005 M magnesium sulphate, 0.00025 M potassium chloride, 0.00025 M calcium chloride and 0.00005 M monopotassium phosphate (low Pi) unless different concentrations are specified. A17 plants were harvested at 8 weeks after inoculation (WAI) for microscopy and quantitative phenotype analysis.

### 4.2. Promoter GUS Reporter Analysis

The inferred *MtbHLHm1;1* promoter, 2038 bp upstream of the bHLHm1;1 start codon and the *MtAMF1;3* promoters, 2180 bp upstream of the AMF1;3 start codon, were PCR amplified with Phusion High Fidelity Taq polymerase (New England Biolabs, Ipswich, MA, USA) from A17 genomic DNA, using promoter-specific primers (Appendix A). Promoter fragments were directionally cloned into *pENTR^TM^/D-TOPO^®^* (Thermo Fisher Scientific, Waltham, MA, USA) resulting in *pENTR-MtbHLHm1;1_pro_* and *pENTR-MtAMF1;3_pro_*. Cloned promoter fragments were sequenced and then introduced individually into *pKGW-GGRR-C-promoter-GUS* [73] via GATEWAY reactions with LR Clonase II (Thermo Fisher Scientific).

*Medicago* A17 plants were transformed with *MtbHLHm1;1_pro_:GUS* and *MtAMF1;3_pro_:GUS* via *A. rhizogenes* and inoculated with AM fungi as described above. Non-inoculated plants of the same age were used as a control. Roots expressing *MtbHLHm1;1_pro_:GUS* and *MtAMF1;3_pro_:GUS* were collected at 8 WAI and incubated overnight in a GUS buffer: 0.05M phosphate buffer, 0.0005 M potassium ferrocyanide, 0.0005 mol potassium ferricyanide, 0.000005 M EDTA, 1% (*v/v*) Triton100 and 50 mg mL^−1^ 5-bromo-4-chloro-3-indolyl-β-D-glucoronic acid (X-Gluc DIRECT). Stained roots were fixed in 70% (*v/v*) ethanol and sectioned by the vibrotome VT1200S (Leica Biosystems, Nussloch, Germany) at 80 µm thickness or embedded in Technovit 7100 resin (Kulzer, Germany) as described by Deguchi et al. [74] and sectioned by a rotary microtome (Leica RM2255) to 10 µm thick. Images of roots with GUS staining were taken using a Leica DM2500M microscope equipped with a DFC500 camera (Leica, Wetzlar, Germany) and processed using Fiji image analysis software (https://fiji.sc).

### 4.3. Protein Localisation and Confocal Microscopy

*MtbHLHm1;1* and *MtAMF1;3* coding sequences were PCR-amplified from eight-week-old mycorrhizal root cDNA using gene-specific primers (Appendix A). *MtbHLHm1;1* (1062 bp) and *MtAMF1;3* (1590 bp) fragments were cloned into *pCR8/GW/TOPO* (Thermo Fisher Scientific) and further ligated into *UBQ_pro_-pK7WGF2* for N-terminal GFP-X fusion using LR Clonase II (Thermo Fisher Scientific) resulting in *UBQ3_pro_:GFP-MtbHLHm1;1* and *UBQ3_pro_:GFP-MtAMF1;3*.

A17 plants were transformed with UBQ3pro:GFP-MtbHLHm1;1 and UBQ3pro:GFP-MtAMF1;3 were inoculated by AM fungi as described above. Non-inoculated plants of the same age were used as a control. Roots expressing UBQ3pro:GFP-MtbHLHm1;1 and UBQ3pro:GFP-MtAMF1;3 were collected at 8 WAI and analysed by confocal microscopy. Roots were fixed in 1% (*v/v*) of freshly prepared paraformaldehyde in 1× PBS (*v/v*), pH 7.4, for 30 min at 4 °C, then washed 3 times in 1× PBS and embedded in 3% (*w/v*) low melting point Agaros (Sigma-Aldrich, St. Louis, MO, USA) in 1× PBS. Embedded roots were sectioned with a vibrotome VT1200S (Leica Biosystems, Nussloch, Germany) at 100 µm thick and blocked in 3% BSA (20 min room temperature) with following washing in 1× PBS. To label fungi inside roots and inside arbuscule-containing cells, 100 µm Agarose root sections were incubated with Wheat Germ Agglutinin (WGA) antibody linked to a TexasRed^®^ fluorophore (Sigma-Aldrich), prepared and used according to the supplier’s instructions. WGA-TexasRed^®^ labelled root sections were mounted on glass slides and analyses by confocal microscopy. Confocal microscopy was performed using a Pascal confocal laser scanning system attached to an Axiovert microscope (Carl Zeiss, Jena, Germany). The GFP and DsRED/WGA-TexasRed^®^ fluorescence was visualised as follows: GFP (argon laser: excitation, 488 nm; emission, 505/530 nm) and DsRED/WGA-TexasRed^®^ (helium-neon laser: excitation, 555 nm; emission, 632 nm). Images were taken with LSM 5 Pascal software (Zeiss, Version 3) and processed using Fiji image analysis software.

### 4.4. Electromobility Shift Analysis

The proteins used in EMSA were expressed using the pMAL^TM^ Protein Fusion and Purification System (New England Biolabs). In brief, the helix-loop-helix region of *MtbHLHm1;1* (Medtr2g010450, amino acid 130–277) was amplified using Velocity DNA polymerase (Bioline), and two restriction recognition sites (*NcoI* and *SbfI*) were introduced during PCR. Then, the PCR product was excised with *NcoI* and *SbfI* and ligated into a *pMAL-c5X* expression vector to create an N-terminal maltose-binding protein (MBP) fusion protein. After sequence validation, the construct was transferred into the *E. coli* strain NEB express. *MBP-MtbHLHm1;1_130–277_* and *MBP* alone were expressed and purified according to the product manual.

The EMSA was performed using the Dig Gel Shift Kit (Roche, Basel, Switzerland). In brief, the 200 bp *MtAMF1.3* promoter *E-box* probe (Medtr4g092770, −726 to −526) was amplified using Velocity DNA polymerase (Bioline). The purified probe DNA was labelled with digoxigenin and labelling efficiency was tested. Then, 31 fmol labelled probe was used in DNA/protein binding reaction with 1 µg MBP-MtbHLHm1;1_130–277_ fusion protein or 1 µg MBP. After 30 min, the binding reaction was loaded onto a 1-h pre-run 6% *v/v* native polyacrylamide gel containing 10% glycerol. DNA was then transferred onto a positively charged nylon membrane using the Trans-Blot^®^Turbo^TM^ transfer system (Bio-Rad, Hercules, CA, USA) and crosslinked onto the membrane by baking at 120 °C for 30 min. The membrane was then hybridised with anti-digoxigenin-AP and the chemiluminescent signal was detected using a ChemiDoc^TM^ XRS+ imager (Bio-Rad).

### 4.5. RNA Interference and Mycorrhizal Phenotype Analysis

RNA interference (RNAi) fragments were designed for *MtbHLHm1;1*, 253 bp and *MtAMF1;3*, 151 bp and PCR amplified from full-length cDNA using gene-specific primers (Appendix A). The PCR fragments were directionally cloned into *pCR8/GW/TOPO* (Thermo Fisher Scientific), sequenced, and further ligated to the RNAi vector *pK7GWIWG2(II)* [72] resulting in *35S_pro_::MtbHLHm1;1* and *35S_pro_::MtAMF1;3.* Both constructs were introduced into A17 plants via *A. rhizogenes.* An empty vector was used as a control. Transformed plants were inoculated with AM fungi (as above) and analysed at 8 WAI. Non-inoculated plants of the same age were used as a control.

Mycorrhizal phenotypes in RNAi experiments were analysed for two main parameters: Mycorrhizal colonisation in roots (MC%) and Arbuscule abundance in colonised roots (Arb%). Roots were fixed in 10% (*v/v*) potassium hydroxide and stained with 5% (*v/v*) acidified ink solution [52] using a Leica MZ10F modular stereo microscope using the grid intersect method as described in [53]. Arbuscule abundance was evaluated under the microscope according to the stain selections outlined by Trouvelot [75]. With each of the four strains used in the mycorrhizal experiments only arbuscules were identified and counted, coiled structures were not evident when in symbiosis with *Medicago*. Degraded or degenerating arbuscules were scored following descriptions presented by Breuillin et al. [34].

### 4.6. Plant Biomass Measurements

Non-inoculated and inoculated *Medicago* plants were dried overnight at 60 °C and weighed to measure total plant biomass. %MGR was calculated based on the differences in DW between non-inoculated and inoculated *Medicago* plants.

### 4.7. RNA Extraction, cDNA Synthesis and Gene Expression Analysis

The total RNA was extracted using the Spectrum^TM^ Plant Total RNA Kit (Sigma-Aldrich) according to the manufacturer’s instructions. For each tissue and treatment, 200 ng of total RNA was converted to cDNA using the iScript^TM^ cDNA Synthesis Kit (Bio-Rad Laboratories Inc., Hercules, CA, USA) according to the manufacturer’s instructions. Gene expression analysis by RT-qPCR was carried out using the SsoAdvanced^TM^ Universal SYBR^®^ Green Supermix (Bio-Rad Laboratories Inc.) on a Light Cycler 480 (Roche). After initial denaturation, the PCR was run for 40 cycles (95 °C 10 s, 55/60 °C 15 s, and 72 °C 20 s) and analysed by the Light Cycler 480 software (Version 1.5). Primers of each amplified gene are listed in Appendix A. Relative expression levels were determined by the 2^−ΔΔCT^ method [49] using the stably expressed *Medicago ubiquitin 10* [76] as a reference gene.

### 4.8. Yeast Mutant 26972c Complementation and ^14^C-Methylammonium Uptake

*MtbHLHm1;1* and *MtAMF1;3* were ligated to the yeast/*E. coli* shuttle vector, *pYES3* resulting in the designated constructs *GAL_pro_*:*MtbHLHm1;1* and *GAL_pro_*:*MtAMF1;3*. Empty vectors *pYES3* and *GAL_pro_*:*ScAMF1* [36] were used as negative and positive controls. All plasmids were introduced into the *Saccharomyces cerevisiae* strain 26972c (*mep1 mep2Δ MEP3 ura-*) which lacks the functional activity of all three MEP ammonium transport proteins [77]. Positive 26972c transformants were selected on minimal medium agar (2% (*w/v*)) containing 0.17% (*w/v*) Yeast Nitrogen Base (YNB), 0.1% L-proline and 2% (*w/v*) Glucose, pH 6.2. Transformed yeast (empty vector, *ScAMF1*, *MtbHLHm1;1,* and *MtAMF1;3*) were inoculated into 20 mL of liquid minimal media in sterile 100 mL glass flasks and incubated at 28 °C/200 rpm overnight. The yeast cultures were diluted to an OD_600_ of 0.1. For each transformant, a 10-fold dilution series was spotted in 5 μL aliquots onto agar plates with selective media as described. The inoculated plates were incubated for 5 days at 28 °C. Each yeast experiment was repeated at least twice. To quantify yeast growth, the initial yeast cultures were diluted to a uniform OD_600_ of 0.01 and grown in 20 mL liquid selective media (as described) with three biological replicates of each. OD_600_ was measured at 10, 24, 32, 50, and 52 h using a UV-VIS Spectrophotometer (SHIMADZU, Kyoto, Japan).

For the ^14^C-Methylammonium uptake measurement, expression of *MtbHLHm1;1* and *MtAMF1;3* was induced at 28 °C/200 rpm for 16 h in the presence of medium containing 0.005M ammonium sulphate, 2% (*w/v*) galactose, and 0.1% (*w/v*) proline. Empty vector *pYES3* and *GALpro:ScAMF1* [36] were used as negative and positive controls. The ^14^C-methylammonium (Perkin-Elmer, Waltham, MA, USA) uptake was conducted and measured as described in Chiasson et al. [36] except that the uptake was measured after 6 min instead of 10 min as described.

### 4.9. MtbHLHm1;1 and MtAMF1 Homology Search and Analysis of Phylogeny

A protein homology search of the *Medicago* bHLHm1 (Medtr2g010450.1/XP_003593344.2) and AMF1 (Medtr2g010370.1/XP_003593336.2) was conducted via the Phytozome12 and National Centre for Biotechnology Information (NCBI) electronic libraries (https://blast.ncbi.nlm.nih.gov/Blast.cgi and https://phytozome.jgi.doe.gov/pz/portal.html (accessed between June–July 2018). MtbHLHm1;1 and MtAMF1 orthologues groups were identified based on maximal similarity of amino acid sequences and protein domain structures. Corresponding protein accessions are listed in Appendix A (Appendix A). Analysis of phylogeny for MtbHLHm1;1 and MtAMF1 was carried out using the bootstrapping procedure at http://phylogeny.lirmm.fr/phylo_cgi/phylogeny.cgi (accessed between June–July 2018). Computed phylogenetic trees were analysed and processed using FigTree v1.4.3 software.

### 4.10. Statistical Analysis

Data in all experiments were compared by either a *t*-test or two-way ANOVA analysis followed by multiple comparison tests as described in the Figure legends using Prism Software (Version 10). Quantified data points represent the mean ± SD.

### 4.11. Accession Numbers

Sequence data used in this work can be found in the Phytozome12/National Centre for Biotechnology Information (NCBI) electronic libraries under accession numbers listed in Appendix A.

## 5. Conclusions

In summary, the data indicates that both MtbHLHm1;1 and MtAMF1;3 have a role in supporting the AM fungal symbiosis in *Medicago*. Their activity may be linked to the management of ammonium inside arbuscule-containing cortical cells, a process that may facilitate the transfer of ammonium from the arbuscule to the plant (Schematic model; Appendix A). This potential transport activity is supported by the observation that MtAMF1;3 can transport methylammonium when expressed in a yeast heterologous expression system. The activity of MtAMF1;3 is also linked with the expression of *MtbHLHm1;1*, a homolog of the soybean nodule-enhanced TF GmbHLHm1, a protein localised to both the symbiosome membrane and infected cell nuclei. Collectively, this change in MtAMF1;3 activity may have significant consequences for NH_4_^+^ management in colonised root cells and may have contributed to observed changes in AM fungal development in colonised roots. Further work is required to understand the role that ammonium plays in the maintenance of arbuscule-colonised cells to support nutritional opportunities or, alternatively, the mitigation of potential ammonium-induced toxicities inside ammonium exporting arbuscule-containing cortical cells.

## Figures and Tables

**Figure 1 ijms-24-14263-f001:**
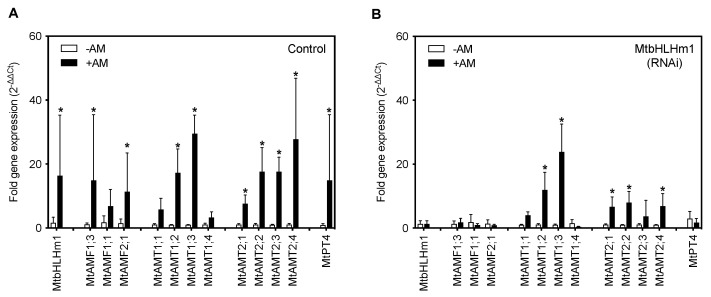
MtbHLHm1;1 regulates gene expression of ammonium and phosphorus transport pathways in response to AM colonisation. (**A**,**B**) Gene expression analysis by RT-qPCR. Values indicate fold-change in expression relative to *UBQ10* [49]. (**A**) Expression in empty vector (Control) roots, (**B**) *35S_pro_::MtbHLHm1;1* (RNAi) roots. Data in (**A**,**B**) represent the mean ± SD (n = 5–10 individual plants). Significance was determined using a two-way ANOVA with a multiple comparison test (Sidak). * indicates significance between −AM and +AM colonised roots (*p* < 0.05).

**Figure 2 ijms-24-14263-f002:**
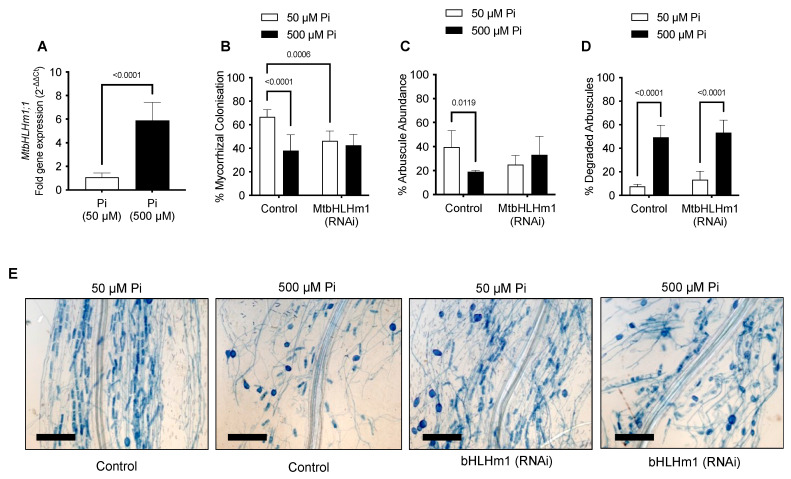
MtbHLHm1;1 influences mycorrhizal colonisation in a Pi-dependent manner. (**A**) Gene expression of *MtbHLHm1;1* in transgenic (empty vector) non-inoculated roots at two Pi (phosphate) levels: 50 and 500 µM. (**A**) Gene expression analysis by RT-qPCR. Values indicate fold-change in expression relative to *UBQ10* [49]. (**B**–**D**) AM colonisation response (%MC) in mycorrhizal roots of empty vector control (Control) and *35S_pro_::MtbHLHm1;1* (RNAi) grown at 50 and 500 µM Pi. (**B**) Mycorrhizal colonisation (%) in colonised roots (MC%); (**C**) Arbuscule abundance (%) in colonised roots; (**D**) Degraded arbuscules (%) in colonised roots. (**E**) Representative mycorrhizal empty vector control (Control) and *35S_pro_::MtbHLHm1;1* (RNAi) roots stained with 5% ink grown at two Pi levels 50 and 500 µM. Scale bars (**E**): 200 µm. Data in (**A**–**D**) represent the mean ± SEM. Significance (*p* < 0.05, *p* < 0.001) was determined using a *t*-test (**A**, n = 12 plants) or two-way ANOVA (**B**–**D**, n = 5–8 plants) with a multiple comparison test (Sidak).

**Figure 3 ijms-24-14263-f003:**
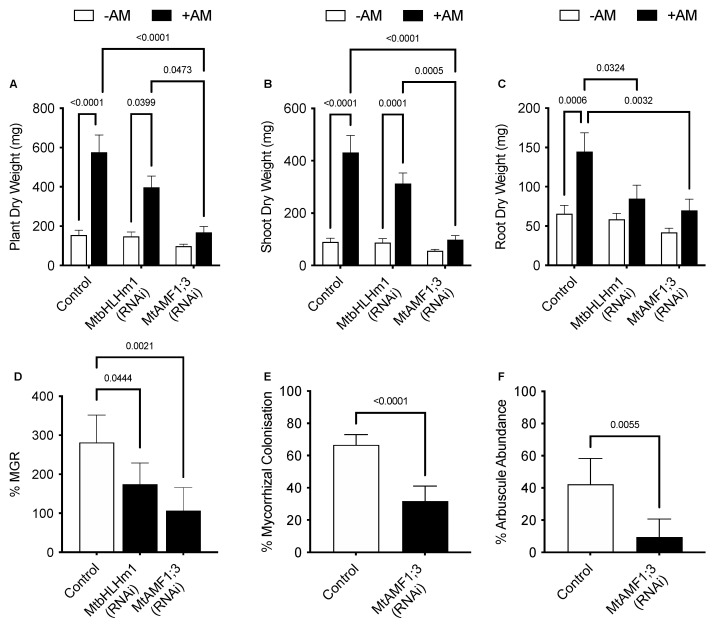
RNAi silencing of *MtbHLHm1;1* and *MtAMF1;3* disrupts growth and arbuscular mycorrhizal colonisation. Quantitative mycorrhizal data of transformed empty vector (Control), *35S_pro_::MtbHLHm1;1* (RNAi) and *35S_pro_::MtAMF1;3* (RNAi) events. (**A**) Biomass per plant (dry weight, mg) of ±AM colonised plants; (**B**) Shoot DW (mg), (**C**) Root DW (mg), (**D**) Mycorrhizal growth response (%) measured in DW growth response between non-inoculated and AM colonised plants, (**E**) Mycorrhizal colonisation (%MC) in AM colonised roots; (**F**) Arbuscule abundance (%Arb) in AM colonised roots. Data in (**A**–**F**) represent the mean ± SEM. Significance (*p* < 0.05, *p* < 0.001) was determined using either two-way ANOVA with a multiple comparison test (Sidak) or an unpaired *t*-test (n = 5–8 transgenic plants).

**Figure 4 ijms-24-14263-f004:**
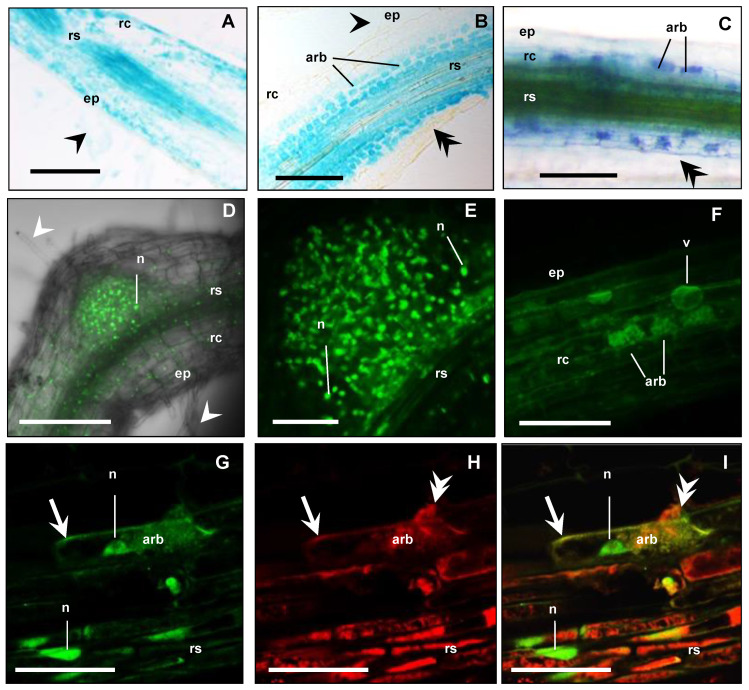
*MtbHLHm1;1* is expressed in non-inoculated and mycorrhizal roots. (**A**–**C**) Promoter GUS expression analysis (blue) of *Medicago* roots expressing *MtbHLHm1;1_pro_:GUS* (8 weeks after inoculation—WAI): (**A**) Non-mycorrhizal root; (**B**,**C**) Mycorrhizal roots; (**A**,**B**) 10 µm resin sections; (**C**) Whole mycorrhizal root stained with 5% ink (dark blue) to visualise fungal structures (hyphae, arbuscules, vesicles). (**D**–**I**) Confocal microscope images of GFP localisation (green channel) in nuclei of root cells expressing *UBQ3_pro_:GFP-MtbHLHm1;1* (8 WAI): (**D**,**E**) non-inoculated root; (**F**–**I**) Mycorrhizal root; (**F**) Root overview, (**G**–**I**) 80 µm vibrotome root section, (**G**) GFP (green channel view, (**H**) Red channel view—DsRED1 autofluorescence and WGA-TexasRed^®^ antibody labelling of AM fungi cell walls; (**I**) Merged channels of (**G**,**H**). On all images: ep = epidermis, rc = root cortex, rs = root stele, arb = arbuscule cells, v = vacuole; white arrowhead shows root hairs, double arrowhead indicates fungal hyphae, n = nucleus, arrow shows cell membranes. Scale bars: (**A**–**D**,**F**), 100 µm; (**E**,**G**–**I**), 20 µm.

**Figure 5 ijms-24-14263-f005:**
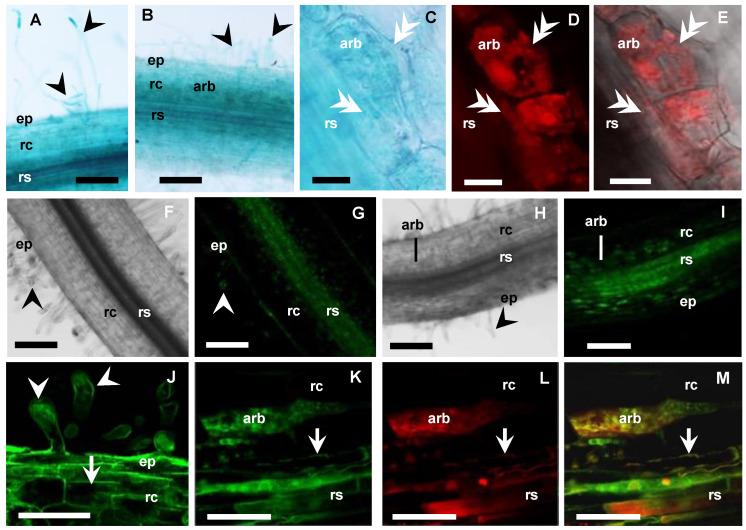
*MtAMF1;3* expressions in non-inoculated and mycorrhizal roots. (**A**–**E**) Medicago roots expressing *MtAMF1;3_pro_:GUS* (8 WAI, blue): (**A**) Non-inoculated root; (**B**–**E**) Mycorrhizal root; (**C**–**E**) Arbuscule-containing cell (80 µm vibrotome root sections); (**C**) Bright field view with GUS expression; (**D**,**E**) AM fungi inside cell, labelled with WGA-Texas Red antibody (excitation 555 nm, red channel). (**F**–**M**) Confocal microscope images of Medicago roots expressing *UBQ3_pro_:GFP-MtAMF1;3* (8 WAI, green channel): (**F**) (DIC), (**G**,**J**) Non-inoculated root; (**H**) (DIC), (**I**,**K**–**M**) Mycorrhizal root; (**F**–**I**) Root overview, (**J**–**M**) Higher magnification of root cells; (**J**) GFP-MtAMF1;3 in root hairs and cortical cells of the Non-inoculated root; (**K**–**M**) GFP-MtAMF1;3 in arbuscule-containing cells of the mycorrhizal root: (**K**) GFP (green channel) view; (**L**) Red channel view—DsRED1 autofluorescence and WGA-TexasRed^®^ antibody labelling of AM fungi cell walls; (**M**) Merged channels. On all images: ep = epidermis, rc = root cortex, rs = root stele, arb = arbuscule cells, black arrowhead shows root hairs, white double arrowhead indicates fungal hyphae, white arrow shows cell membranes. Scale bars: (**A**,**B**,**F**–**I**) 100 µm; (**C**–**E**,**J**–**M**) 20 µm.

**Figure 6 ijms-24-14263-f006:**
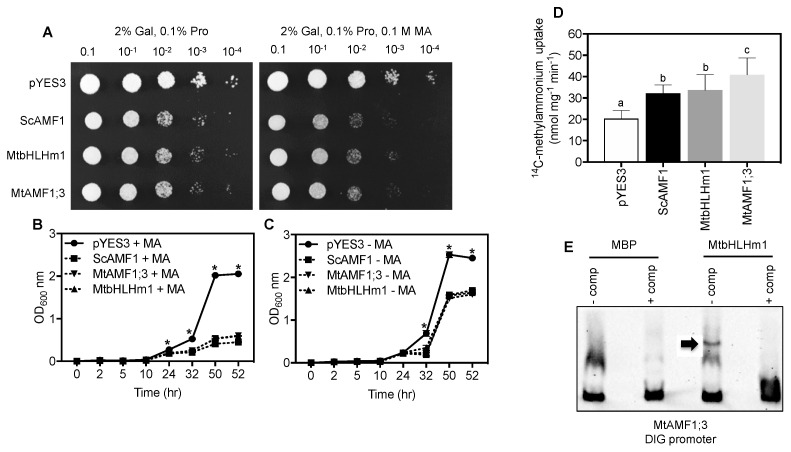
MtbHLHm1;1 and MtAMF1;3 enhance methylammonium (MA) transport in the *Saccharomyces cerevisiae* ammonium transport mutant 26972c (*mep1 mep2Δ* MEP3). (**A**) Serial dilutions of yeast growth on agar plates containing 0.1% L-Proline ± 0.1 M MA. The lack of an effective MEP ammonium transport system allows 26972c to grow on toxic concentrations of methylammonium (0.1 M MA) when transformed with an empty vector (pYES3). Yeast AMF1 (ScAMF1, Chiasson et al. 2014) was used as a positive control. Note: Reduced growth of 26972c on 0.1 M MA when expressing either *MtbHLHm1;1* or *MtAMF1;3* similar to the positive control *ScAMF1*. (**B**,**C**) Liquid, yeast growth curves (n = 3, biological replicates) measured as change in absorbance at OD_600_ in the presence (**B**) or absence (**C**) of 0.1 M MA. (**D**) Unidirectional influx (6 min) of ^14^C-methylammonium (1 mM) into yeast cells (n = 10 biological replicates). (**E**) Electromobility shift analysis of a truncated DIG-labelled *MtAMF1;3* promoter by a purified MPB-MtbHLHm1;1 fusion probe. DIG-labelled DNA was mixed with ±competitor (125-fold unlabelled *MtAMF1;3* promoter DNA) and 1 µg MPB-MtbHLHm1;1. The arrow indicates the gel-shift product. Data in (**B**–**D**) represent the mean ± SD (n = 10 and 3 biological replicates, respectively). Significance was determined using a one (**D**) or two-way (**B**,**C**) ANOVA with multiple comparison tests (Tukeys and Dunnets, respectively). Different letters or * indicate significance (*p* < 0.05).

## Data Availability

The data presented in this study are available on request from the corresponding author.

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
