# Peer review of "Arbuscular-Mycorrhizal Symbiosis in Medicago Regulated by the Transcription Factor MtbHLHm1;1 and the Ammonium Facilitator Protein MtAMF1;3"

_ijms, 2023, doi:10.3390/ijms241814263_

Round 1
Reviewer 1 Report
The manuscript by Ovchinnikova and colleagues reports a role for the Medicago bHLHm1 transcription factor and one of its transcriptionally regulated target genes MtAMF1;3, encoding an presumed ammonium transporter, in the regulation of AM symbiosis. Little is known about the regulation of ammonium transport (or N in general) in the symbiosis with AM fungi. Therefore, this work offers a novel insight and entry point into unravelling this important but complex aspect of the symbiosis. Furthermore, it suggests a regulatory role for this module in the both the AM and rhizobium symbioses.
However, several point need more clarification:
1. In the introduction CBX1 is mentioned. It would be good to also include reference to WRI5 in Medicago (Jiang et al., 2018).
2. The role of AMT2 transporters in the control of arbuscule development/degradation (reference 30) should be better described.
3. Was there a particular reason why the work was done in Medicago rather than in soybean where GmbHLHm1 was first studied?
4. Add info on the phylogenetic analyses in the materials and methods section.
5. Throughout the manuscript gene names, constructs and full species names should be italic.
6. Remove “AM enhanced” from line 170. I guess the reduction is not AM dependent…
7. It is mentioned that MtAMT2;4 expression is reduced in the bhlhm1 RNAi roots, but from Fig. 1B it still seems to be induced by AM. A proper control to substantiate this claim is missing.
On the other hand it seems that induction of MtAMF2;1 is impaired upon silencing of bHLHm1.
8. Is the induction of MtbHLHm1 expression upon exposure of colonized roots to 500 µM Pi dependent of mycorrhization? In other words is the expression of bHLHm1 itself nutrient dependent in the absence of AMF?
9. Line 210; here it is mentioned that the method by McGonigle is used for scoring the mycorrhization, while in the material and methods Trouvelot et al. is mentioned. Please clarify. Also describe how degraded arbuscules were scored. Potentially close-up images could be added in the main Fig 2E?
10. Explain %mycorrhiza growth response
11. At line 225 it is mentioned that %Arb was only reduced in the MtAMF1;3 RNAi roots. However, in Fig. 2C there is also a significant reduction upon silencing of MtbHLHm1. Please clarify.
12. For both MtbHLHm1 and MtAMF1;3 RNAi, do these constructs lead to cross-silencing of close homologs?
13. I find the expression analyses of both MtbHLHm1 and MtAMF1;3 not very convincing. In Fig. 4A and 5A different regions of the roots seem to be shown. How is the expression of these genes in relation to root development? Is it nutrient dependent? Is there an upregulation in arbuscules (the presented figures are not very clear)? Does its expression depend on arbuscule developmental stage. Is there any support for the expression profiles from publically available transcriptome data, for example the Medicago gene atlas server (https://lipm-browsers.toulouse.inra.fr/pub/expressionAtlas/app/)?
14. Also the images of the subcellular localizations are not very clear. Membrane localization is claimed for MtAMF1;3 and possibly MtbHLHm1 but I don’t see any data that would support these claims.
15. How was the WGA staining done..details are lacking in the M&M.
16. Line 349; I am not sure co-localization is shown..maybe phrase differently.
17. Lines 381-387; does this mean that the subcellular localization of MtAMF1;3 in arbusculated cells was already reported, or was it shown for other members of this family? At line 388 a comparison with LjAMT2;2 is made..is this the ortholog of MtAMF1;3?..the suggestion that MtAMF1;3 has a role outside of arbusculated cells seems better supported by the promoter studies.
18. At line 391 Gaude et al. is mentioned, but perhaps it would be better to more extensively examine the expression of the genes in publically available data (as mentioned earlier the expression atlas, but also for example Hogekamp and Kuster, BMC Genomics 2013 or Zeng et al. Plant J. 2018.
19. Line 397; this conclusion is not well supported by the images.
20. Line 399/400: can the authors better explain how they envision that bHLHm1 silencing can increase colonization levels, while at the same time reducing the amount of (healthy) arbusucles?
21. Line 403, only MtAMT2;3 has been linked to maintenance of arbuscules. Furthermore, AMT2;3 did not appear to be transporting ammonium, but was suggested to maybe have a signaling role.
22. Line 406 – 417: I do not see the relevance of this part of the text.
23. At line 430 in vitro affinity assays are mentioned, but these experiments are not mentioned in the results. Or does this refer to the EMSA, in which case the sentence should be formulated differently.
24. Line 436: as mentioned earlier a role for MtbHLHm1 in the regulation of MtAMT2 expression is not clearly shown.
25. Line 452: the role for bHLHm1 as a Pi-mediated repressor is not clear to me.
26. The same holds for the discussion at lines 459-470 with respect to “such a mechanism”at line 470. Please explain.
27. Please explain the presented model (Suppl Fig. 3) better in the text.
Author Response
Response to Reviewer Comments
Reviewer 1
- In the introduction CBX1 is mentioned. It would be good to also include reference to WRI5 in Medicago (Jiang et al., 2018).
Response 1: We have included reference to WR15a (Jiang et al., 2018) at lines 52-54. Great suggestion and one we shouldn’t have missed.
- The role of AMT2 transporters in the control of arbuscule development/degradation (reference 30) should be better described.
Response 2: We have expanded our interpretation of AMT2 activity Lines 106-111 – ‘The evidence for a functional transport link between the AM symbiosis and AMT2 transporters at the periarbuscular membrane is still unclear. Breuillin-Sessoms et al. [1] showed that N starvation to be an effective suppressor of premature arbuscule degradation (PAD) commonly observed in Mt-pt4 mutants. AMT2.3 activity was required for PAD suppression but loss of amt2.3 had no influence on shoot N content when in symbiosis.
- Was there a particular reason why the work was done in Medicago rather than in soybean where GmbHLHm1 was first studied?
Response 3: We conducted preliminary experiments in soybean to test the response between GmbHLHm1 and AM colonisation, which showed a similar response with a disruption in AM colonisation. However, at the time we were actively transitioning to a Medicago plant model to study bHLHm1 and AMF3 activities. This work continued in Medicago thereafter.
- Add info on the phylogenetic analyses in the materials and methods section.
Response 4: We have included the phylogenetic analysis in the M&M section (line 673-683)
“4.9. MtbHLHm1 and MtAMF1 Homology Search and Analysis of Phylogeny
Protein homology search of the Medicago bHLHm1 (Medtr2g010450.1/ XP_003593344.2) and AMF1 (Medtr2g010370.1/XP_003593336.2) was conducted via the Phytozome12 and National Centre for Biotechnology Information (NCBI) electronic libraries (https://blast.ncbi.nlm.nih.gov/Blast.cgi and https://phytozome.jgi.doe.gov/pz/portal.html. MtbHLHm1 and MtAMF1 orthologues groups were identified based of maximal similarity of amino acid sequences and protein domain structures. Corresponding protein accessions are listed in Supplementary Material (Supplementary Tables 1 and 2). Analysis of phylogeny for MtbHLHm1 and MtAMF1 was done using bootstrapping procedure at http://phylogeny.lirmm.fr/phylo_cgi/phylogeny.cgi. Computed phylogenetic trees were analysed and processed using FigTree v1.4.3 software.”
- Throughout the manuscript gene names, constructs and full species names should be
Response 5: We have evaluated the entire manuscript and corrected the italics for gene names, constructs and full species names.
- Remove “AM enhanced” from line 170. I guess the reduction is not AM dependent
Response 6: The AM dependent comment was removed.
- It is mentioned thatMtAMT2;4 expression is reduced in the bhlhm1 RNAi roots, but from Fig. 1B it still seems to be induced by AM. A proper control to substantiate this claim is missing.
Response 7: Apologies for the confusion with this description and not being clearer on the result. MtAMT2;4 expression remains AM enhanced with the RNAi loss of Mtbhlhm1. The relative level of expression (Fold change) to the control plants suggests expression is lower. Unfortunately, AtAMT 2.3 instead of AtAMT2;4 was identified as having enhanced expression relative to the -AM controls in the RNAi background. We have modified this statement to reflect the observed data and correctly identify AtAMT2;4.
“The reduction in Mtbhlhm1;1 expression also enhanced the AM-induced expression of MtAMT1;2, MtAMT2;1, MtAMT2;2 and MtAMT2;4 relative to the -AM controls (Figure 1B, P<0.05). The loss of AM enhanced expression of MtAMF1;3 and the ammonium transporter MtAMT2;3 [1] in Mtbhlhm1;1 RNAi silenced roots suggests a potential role of MtbHLHm1 in the molecular control of two ammonium transport pathways in mycorrhizal colonised roots.”
- On the other hand it seems that induction ofMtAMF2;1 is impaired upon silencing of bHLHm1.
Response 8: We have included reference to MtAMF2;1, thank you for identifying that detail we failed to elaborate on. Changes reflecting this are contained in lines 161-163 and 180-185
“MtbHLHm1;1, MtAMF1;3 and MtAMF2;1 expression were significantly (P<0.05) enhanced when inoculated with AM fungi (Figure 1A).”
“The loss of AM enhanced expression of MtAMF1;3, MtAMF2;1 and the ammonium transporter MtAMT2;3 [1] in Mtbhlhm1;1 RNAi silenced roots suggests a potential role of MtbHLHm1;1 in the molecular control of multiple ammonium transport pathways in mycorrhizal colonised roots?”
- Is the induction of MtbHLHm1 expression upon exposure of colonized roots to 500µM Pi dependent of mycorrhization? In other words is the expression of bHLHm1 itself nutrient dependent in the absence of AMF?
Response 9: In Figure 2A, we show that Pi (500µM) increased the expression of MtbHLHm1;1 in non-AM plants. Yes, the expression of MtbHLHm1;1 responds to Pi supply. See clarification Lines 201-203
“We found MtbHLHm1 expression was significantly enhanced (~6 fold, p<0.05) in WT roots non AM roots exposed to high Pi (500 µM) relative to those grown at low Pi (50 µM) (Figure 2A).”
- Line 210; here it is mentioned that the method by McGonigle is used for scoring the mycorrhization, while in the material and methods Trouvelot et al. is mentioned. Please clarify. Also describe how degraded arbuscules were scored. Potentially close-up images could be added in the main Fig 2E?
Response 10: The McGonigle et al method refers only to the use of the GRID INTERSECT method they developed to standardise the counting of arbuscules relative to root length. Trouvelot et al refers to the staining procedures to better define arbuscule structures from coiled structures using 0.1% carbol fuchsin. Degraded Arbuscules were scored following the same methods to score mature arbuscules but followed the physical descriptions of degraded arbuscules outlined in Breuuillin-Sessoms et al Plant Cell 2015. We have modified the description of the methods (see lines ??).
“Roots were fixed in 10% (v/v) potassium hydroxide and stained with 5% (v/v) acidified ink solution [2] using a Leica MZ10F modular stereo microscope using the grid intersect method as described in [3]. Arbuscule abundance was evaluated under the microscope according to the stain selections outlined by Trouvelot [4]. With each of the four strains used in the mycorrhizal experiments only arbuscules were identified and counted, coiled structures were not evident when in symbiosis with Medicago. Degraded or degenerating arbuscules were scored following descriptions presented by Breuillin et al [1].”
- Explain %mycorrhiza growth response
Response 11: The growth of the plants ± AM inoculation was compared. %mycorrhiza growth response is the % difference in growth between mock and AM inoculated plants at harvest. Plant tissues (whole plants) were dried at 60°C overnight and weighed. This detail was added to the materials and methods section (line 640-641) and in the legend of Figure 3B (Line 240-241)
“Non-inoculated and inoculated Medicago plants were dried overnight at 60°C and weighed to measure total plant biomass. %MGR was calculated based on the differences in DW between non-inoculated and inoculated Medicago plants.”
“Figure 3. RNAi silencing of MtbHLHm1 and MtAMF1;3 disrupts growth and arbuscular mycorrhizal colonisation. Quantitative mycorrhizal data of transformed empty vector (Control), 35Spro::Mtbhlhm1;1 (RNAi) and 35Spro::Mtamf1;3 (RNAi) events. A, Biomass per plant (dry weight, mg) of ± AM colonised plants; B, Shoot DW (mg), C, Root DW (mg), D, Mycorrhizal growth response (%MGR) measured in DW growth response between non-inoculated and AM colonized plants, E, Mycorrhizal colonization (%MC) in AM colonised roots; F, Arbuscule abundance (%Arb) in AM colonised roots. Data in A-F represent the mean ± SEM. Significance (P<0.05, P<0.001) was determined using either Two-way ANOVA with a multiple comparison test (Sidak) or an unpaired t-test (n=5-8 transgenic plants).”
- At line 225 it is mentioned that %Arb was only reduced in the MtAMF1;3 RNAi roots. However, in Fig. 2C there is also a significant reduction upon silencing ofMtbHLHm1. Please clarify.
Response 12: We have also carefully reviewed this data. We have incorporated a larger data set taken from multiple experiments to better estimate changes in measured %MC and %Arb (Figure 2B and 2C). We have also included the impact of +AM on plant growth, shoot and root dry weights (Figure 3). Together with the revised analysis the data shows that %Arb doesn’t decrease in the Mtbhlhm1;1 plants relative to the controls (Figure 2C). However, the revised data also showed a significant reduction in %MC (Figure 2B) in the Mtbhlhm1;1 line (Figure 2B), which was similar to both low (50 µM) and high (500 µM) Pi concentrations. We have reformatted Figures 2 and 3 to simplify the data presentation and removed redundant data (Mtbhlhm1;1 %MC and %Arb, that was previously located in Figure 3.
We apologise for this confusion but welcome the increased robustness of the data with the larger data sets being analysed. Text reflecting this change can be found in lines 224-241.
“Plant and shoot dry weights increased with AM inoculation in the Mthlhm1;1 RNAi line; though root growth remained similar to the non-inoculated plants (Figure 3A-C). Collectively, the mycorrhizal growth response (%MGR) was significantly lower in the Mtbhlhm1;1 RNAi line (Figure 3D). The AM fungal structures in the Mtbhlhm1;1 RNAi line were visualised using the ink staining method according to [2] and scored (Figure 2B-D) using the grid intersect method as described in [3]. Roots were evaluated for total mycorrhizal colonisation (%MC, which includes hyphae, arbuscules and vesicles) and arbuscule abundance (%Arb) (Figure 2B, 2C). In general, control plants showed a significant reduction in %MC and %Arb when supplied 500 µM Pi (Figure 2B, 2C). The loss of Mtbhlhm1;1 significantly decreased %MC (p=0.0006), though the impact was similar at low and high Pi treatments (Figure 2B). We found no significant change in %Arb between the control and Mtbhlhm1;1 at either Pi concentration (Figure 2C). %Arbuscule degradation increased in both the empty vector control and the Mtbhlhm1 roots in response to elevated Pi (500 µM) (Figure 2D).
We then developed knockdown lines for MtAMF1;3 (35Spro::Mtamf1;3) using RNAi (Supp Figure 2). In the presence of AM fungi and 50 µM Pi, empty vector (control) transformed plants responded to AM colonisation through increased plant, shoot and root dry weights (Figure 3A-C). The AM enhanced growth was reduced with RNAi silencing of Mtamf1;3 (35Spro::Mtamf1;3) lines when presented as either total plant dry weight, shoot dry weight and root dry weight (Figure 3A-C). %MGR, %MC and %Arb all decreased significantly to the control lines (Figure 3D-F, respectively). Ink-stained overviews of AM colonised roots of Mtamf1;3 show a general reduction in colonisation and arbuscule presentation relative to empty vector controls (Supp Figure 2C,E).”
- For both MtbHLHm1 and MtAMF1;3 RNAi, do these constructs lead to cross-silencing of close homologs?
Response 13: We previously conducted experiments to test the sensitivity of the RNAi constructs. We found the 35Spro::Mtbhlhm1;1 (RNAi) construct disrupted +AM induction of MtbHLHm1;1, MtbHLHm1;2, MtAMF1;3 and a close homolog, MtAMF1;1. We created a 35Spro::Mtbhlhm1;2 (RNAi) construct which reduced the +AM induction of MtbHLHm1, MtbHLHm2, MtAMF1;1 and MtAMF1;3 expression. 35Spro::Mtbamf1;3 (RNAi) was only specific to MtAMF1;3. We have included this preliminary data in the Supplementary Figure 3 in the revised manuscript. Commentary to this preliminary work can be found at lines 180-187
“The specificity of 35Spro::Mtbhlhm1;1 was also tested against other genes including its close homolog MtbHLHm1;2 and a MtAMF1;3 homolog, MtAMF1;1 (Supp Figure 3). 35Spro::Mtbhlhm1;1 also silenced MtbHLHm1;2 and MtAMF1;1. A control construct, 35Spro::Mtbhlhm2, was also developed and tested for its impact on MtbHLHm1;1 and MtAMF1;3 expression. Mtbhlhm2 reduced the scale of the +AM response for MtbHLHm1;1, MtbHLHm1;2, and MtAMF1;1 expression, while eliminating all +AM responsiveness in MtAMF1;3 (Supp Figure 3). We can conclude that MtbHLHm1;1 and MtbHLHm1;2 have some redundancy and both influence the expression of MtAMF1;3.”
- I find the expression analyses of bothMtbHLHm1 and MtAMF1;3 not very convincing. In Fig. 4A and 5A different regions of the roots seem to be shown. How is the expression of these genes in relation to root development? Is it nutrient dependent? Is there an upregulation in arbuscules (the presented figures are not very clear)? Does its expression depend on arbuscule developmental stage. Is there any support for the expression profiles from publically available transcriptome data, for example the Medicago gene atlas server (https://lipm-browsers.toulouse.inra.fr/pub/expressionAtlas/app/)?
- Also the images of the subcellular localizations are not very clear. Membrane localization is claimed for MtAMF1;3 and possibly MtbHLHm1 but I don’t see any data that would support these claims.
Response 14 & 15: We appreciate the comments and attention to the qualitative images showing cellular expression of MtbHLHm1;1 (Figure 4) and MtAMF1;3 (Figure 5). We completed these experiments to highlight cellular/tissue expression in ±AM inoculated Medicago roots. In this study, we didn’t explore the relationship to root development or tested for changes in response to nutrient provision, outside growing the plants at low (50 µM) or high (500 µM) exogenous Pi (Figure 2). This type of study was limited by the fact we were using a hairy root system which limited the number of transgenic events we could generate and examine. This of course will be examined in our future work as we are in the process of collecting stable knockdown mutants for Mtbhlhm1;1 and Mtamf1;3.
We believe the images in Figures 4 and 5 show the localisation of the GUS signal in line with the localisation of GFP tagged protein. +AM inoculation demonstrates a strong induction of GUS signal in cells containing arbuscules. This result indicates a developmental stage of the symbiosis where MtbHLHm1 responds, similar to that of GmbHLHm1, which shows a nodule enhanced expression pattern when in symbiosis with rhizobia bacteria. In non-inoculated roots, MtAMF1;3 is diffusely expressed across the majority of cellular tissues, but then targeted to membranes when tagged with GFP (Figure 5J).
For this study we didn’t have suitable antibodies to either MtbHLHm1;1 or MtAMF1;3 to conclusively localise both proteins to specific membrane structures and or organelles. The in-situ overexpression of GFP tagged protein does highlight a default targeting of MtbHLHm1 protein to the nucleus as well as to cells containing arbuscules and highlighting the arbuscule structure (Fig 4F). MtbHLHm1;1 is assumed to how similar targeting to membranes based on its c-terminus hydrophobic tail and its strong amino acid similarity to GmbHLHm1 which is a membrane targeted protein as well as one directed to the nucleus. For MtAMF1;3, the localisation of the GFP tagged membrane protein is much more prominent ±AM inoculation. Arbuscule structures are clearly targeted (Figure 5K).
- How was the WGA staining done..details are lacking in the M&M.
Response 16: The description of the WGA staining was presented in the materials and methods. We have included a reference (Harrison et al [5]) we used to develop the protocol. See lines 606-608.
“Roots expressing UBQ3pro:GFP-MtbHLHm1;1 and UBQ3pro:GFP-MtAMF1;3 were collected at 8 WAI and analysed by confocal microscopy. To label fungi inside roots and inside arbuscule-containing cells, mycorrhizal root sections were incubated with Wheat Germ Agglutinin (WGA) antibody linked to a TexasRed® fluorophore (Sigma-Aldrich) as described by Harrison et al [5]. Confocal microscopy was performed using a Pascal confocal laser scanning system attached to an Axiovert microscope (Carl Zeiss, Germany). The GFP and DsRED/WGA-TexasRed® fluorescence was visualised as follows: GFP (argon laser: excitation, 488 nm; emission, 505/530 nm) and DsRED/WGA-TexasRed® (helium-neon laser: excitation, 555 nm; emission, 632 nm). Images were taken with LSM 5 Pascal software (Zeiss) and processed using Fiji image analysis software.”
- Line 349; I am not sure co-localization is shown..maybe phrase differently.
Response 17: We have re-written this sentence to reflect the context of the sentence (Line 372-375).
“Overexpressed GFP-tagged MtAMF1;3 protein was also located with MtbHLHm1;1 in arbuscule containing cortical cells, while electromobility shift analysis revealed the potential for direct transcriptional interaction between MtbHLHm1;1 and the promoter of MtAMF1;3.”
- Lines 381-387; does this mean that the subcellular localization of MtAMF1;3 in arbusculated cells was already reported, or was it shown for other members of this family? At line 388 a comparison with LjAMT2;2 is made..is this the ortholog of MtAMF1;3?..the suggestion that MtAMF1;3 has a role outside of arbusculated cells seems better supported by the promoter studies.
Response 18: This is the first description of MtbHLHm1 and MtAMF1;3 in Medicago and in the context of arbuscular mycorrhizal interactions. LjAMT2;2 is a member of the high-affinity ammonium transporter family with no sequence homology to AMF transport proteins – quite different proteins and activities. We highlight the similarity in the cellular expression profiles between LjAMT2;2 and MtAMF1;3 ±AM inoculation in these two different legume species (Lotus and Medicago). We assume MtAMF1;3 has multiple roles in plant cells ± AM inoculation. The +AM inoculation results in a concentration of MtAMF1;3 in cells where arbuscules are found.
- At line 391 Gaude et al. is mentioned, but perhaps it would be better to more extensively examine the expression of the genes in publically available data (as mentioned earlier the expression atlas, but also for example Hogekamp and Kuster, BMC Genomics 2013 or Zeng et al. Plant J. 2018.
Response 19: This is a great suggestion for the next study on these two genes. We believe our thorough interrogation of MtbHLHm1;1 and MtAMF1;3 expression in Medicago and the analysis of associated genes known to be important to the AM symbiosis (Pi and N genes) provides sufficient context for the questions we are raising and trying to answer – does MtbHLHm1;1 and MtAMF1;3 have a role in regulating the AM symbiosis in Medicago? A more detailed whole genome transcriptional analysis has been completed for the soybean homolog GmbHLHm1 in both nodules and root tissues [6]. Insight from this study has led us to follow on in this work.
- Line 397; this conclusion is not well supported by the images.
Response 20: We have refined this section to include (Lines 398-400):
“Using GUS, this change in expression was demonstrated by a refined cellular localisation of promoter driven GUS expression (Arbuscules and the root stele).”
- Line 399/400: can the authors better explain how they envision that bHLHm1 silencing can increase colonization levels, while at the same time reducing the amount of (healthy) arbusucles?
Response 21: We have observed that under elevated levels of supplied Pi (500 µM), colonisation levels in Mtbhlhm1;1 plants were maintained which is opposite to that of control roots (Figure 2B). There was still a strong influence of Pi on arbuscule function as degraded arbuscules increased with 500 µM Pi in both control and Mtbhlhm1;1 plants. The ability of MtbHLHm1;1 to influence %Arb and is most likely through its influence on MtAMF1;3 expression which is strongly repressed in the Mtbhlhm1;1 RNAi lines. Loss of Mtamf1;3 resulted in strong reduction of plant DW, %MGR, %MYC and %Arb.
- Line 403, only MtAMT2;3 has been linked to maintenance of arbuscules. Furthermore, AMT2;3 did not appear to be transporting ammonium, but was suggested to maybe have a signaling role.
Response 22: With the loss of Mtbhlhm1 activity, we observed a fold-change reduction in the expression of MtAMT2;1, 2;2, 2;3 and 2;4 upon +AM inoculation. We can only conclude that the differences between previous AMT2 links to arbuscule maintenance must be due to the change in MtbHLHm1 expression examined in this study. This study did not explore the functional activities of AMT2 proteins, which was out of scope of the study.
- Line 406 – 417: I do not see the relevance of this part of the text.
Response 23: The highlighted text was provided to stress that multiple cell types (±AM inoculation) express AMF and or AMT genes and that there are other activities of these proteins in the cell. We have re-written the text to help clear this misunderstanding:
“However, there are differences, such as the localization of GFP-tagged MtAMF1;3 protein at the plasma membrane of root hairs and cortical cells of non-mycorrhizal roots, an outcome mostlikely linked to the MtAMF1;3 protein not being under the control of its own promoter. Work by Guether et al [7] also demonstrated using RT-PCR analysis, that LjAMT2;2 was expressed in not only arbusculated cells but also in non-colonized cortical cells from mycorrhizal roots and cortical cells from nonmycorrhizal roots. This suggested that there is a possible role for MtAMF1;3 in other root cell types other than arbusculated cortical cells in Medicago.”
- At line 430 in vitro affinity assays are mentioned, but these experiments are not mentioned in the results. Or does this refer to the EMSA, in which case the sentence should be formulated differently.
Response 24: We have clarified the work in this sentence – this was work completed by other using MtbHLH2:
“Further, independent analysis by chromatin immunoprecipation demonstrated that a separate MtbHLH2 bound directly to the promoter of MtCP77 to inhibit its expression [8].”
- Line 436: as mentioned earlier a role for MtbHLHm1 in the regulation of MtAMT2 expression is not clearly shown.
Response 25: We have shown in Figure 1B, a clear down regulation (fold-change) in expression of AMT2 with in the Mtbhlhm1;1 lines (+AM).
- Line 452: the role for bHLHm1 as a Pi-mediated repressor is not clear to me.
- The same holds for the discussion at lines 459-470 with respect to “such a mechanism”at line 470. Please explain
Response 26-27: We have re-written this section to help clarify how MtbHLhm1 and AM symbiosis interacts.
“MtbHLHm1;1 expression is strongly induced by Pi supply in -AM roots (Figure 2A). High expression was linked to a reduction in %MYC and %Arbuscules in control roots (Figure 2B, C). Elevated Pi increased the % of degraded arbuscules. However, the loss of Mtbhlhm1;1 reversed the drop in %MC (Figure 2B) but had no impact on % Arb in AM colonised roots at either 50 or 500 µM Pi (Figure 2C). We saw no change in the abundance of degraded arbuscule structures with 500 µM Pi (Figure 2D). These results with the controls are similar to that observed by many other AM studies where %MC and %Arb are often reduced by increased exposure to high Pi concentrations [9-12]. The Pi independent changes in %MC and %Arb with the loss of Mtbhlhm1;1 suggests a direct relationship between the MtbHLHm1;1 TF and AM development.”
- Please explain the presented model (Suppl Fig. 3) better in the text.
Response 28: We have included a short statement incorporating the model into the discussion (Line: 528-537)
“In summary, we highlight the MtbHLHm1;1 TF in the regulation of the AM symbiosis in Medicago truncatula. MtbHLHm1;1 is a Pi enhanced TF that in response to AM inoculation increases the expression of several genes linked to ammonium transport, including AMT and AMF genes. MtbHLHm1 expression is linked to the activity of the low affinity AMF transporter family, previously observed in soybean root nodules. Loss of Mtamf1;3, compromises AM development in Medicago roots. Cellular expression and protein localization studies identify both MtbHLHm1;1 and MtAMF1;3 are associated with root cells containing arbuscules. We suggest AMF activity influences ammonium transport in AM colonized cells to support the symbiotic partnership (Supplementary Figure 4).”
References:
- Breuillin-Sessoms, F.; Floss, D.S.; Gomez, S.K.; Pumplin, N.; Ding, Y.; Levesque-Tremblay, V.; Noar, R.D.; Daniels, D.A.; Bravo, A.; Eaglesham, J.B.; et al. Suppression of arbuscule degeneration in Medicago truncatula phosphate transporter4 mutants is dependent on the ammonium transporter 2 family protein AMT2;3. The Plant Cell 2015, 27, 1352-1366, doi:10.1105/tpc.114.131144.
- Vierheilig, H.; Coughlan, A.P.; Wyss, U.; Piche, Y. Ink and vinegar, a simple staining technique for arbuscular-mycorrhizal fungi. Appl. Environ. Microbiol. 1998, 64, 5004-5007.
- McGonigle, T.P.; Miller, M., H.; Evans, D., G.; Fairchild, G., L.; Swan, J., A. A new method which gives an objective measure of colonization of roots by vesicular—arbuscular mycorrhizal fungi. New Phytol. 1990, 115, 495-501, doi:10.1111/j.1469-8137.1990.tb00476.x.
- Trouvelot, A. Mesure du taux de mycorhization VA d'un systeme radiculaire. Recherche de methodes d'estimation ayant une significantion fonctionnelle. Mycorrhizae : physiology and genetics 1986, 217-221.
- Harrison, M.J.; Dewbre, G.R.; Liu, J. A phosphate transporter from Medicago truncatula involved in the acquisition of phosphate released by arbuscular mycorrhizal fungi. The Plant Cell 2002, 14, 2413-2429.
- Chiasson, D.M.; Loughlin, P.C.; Mazurkiewicz, D.; Mohammadidehcheshmeh, M.; Fedorova, E.E.; Okamoto, M.; McLean, E.; Glass, A.D.M.; Smith, S.E.; Bisseling, T.; et al. Soybean SAT1 (Symbiotic Ammonium Transporter 1) encodes a bHLH transcription factor involved in nodule growth and NH4+ transport. Proceedings of the National Academy of Sciences 2014, 111, 4814-4819, doi:10.1073/pnas.1312801111.
- Guether, M.; Neuhäuser, B.; Balestrini, R.; Dynowski, M.; Ludewig, U.; Bonfante, P. A mycorrhizal-specific ammonium transporter from Lotus japonicus acquires nitrogen released by arbuscular mycorrhizal fungi. Plant Physiol. 2009, 150, 73-83, doi:10.1104/pp.109.136390.
- Deng, J.; Zhu, F.G.; Liu, J.X.; Zhao, Y.F.; Wen, J.Q.; Wang, T.; Dong, J.L. Transcription Factor bHLH2 Represses CYSTEINE PROTEASE77 to Negatively Regulate Nodule Senescence. Plant Physiology 2019, 181, 1683-1703, doi:10.1104/pp.19.00574.
- Breuillin, F.; Schramm, J.; Hajirezaei, M.; Ahkami, A.; Favre, P.; Druege, U.; Hause, B.; Bucher, M.; Kretzschmar, T.; Bossolini, E.; et al. Phosphate systemically inhibits development of arbuscular mycorrhiza in Petunia hybrida and represses genes involved in mycorrhizal functioning. The Plant Journal 2010, 64, 1002-1017, doi:10.1111/j.1365-313X.2010.04385.x.
- Balzergue, C.; Chabaud, M.; Barker, D.G.; Becard, G.; Rochange, S.F. High phosphate reduces host ability to develop arbuscular mycorrhizal symbiosis without affecting root calcium spiking responses to the fungus. Frontiers in plant science 2013, 4, 426, doi:10.3389/fpls.2013.00426.
- Nouri, E.; Breuillin-Sessoms, F.; Feller, U.; Reinhardt, D. Phosphorus and nitrogen regulate arbuscular mycorrhizal symbiosis in Petunia hybrida. PLOS ONE 2014, 9, e90841, doi:10.1371/journal.pone.0090841.
- Kobae, Y.; Ohmori, Y.; Saito, C.; Yano, K.; Ohtomo, R.; Fujiwara, T. Phosphate treatment strongly inhibits new arbuscule development but not the maintenance of arbuscule in mycorrhizal rice roots. Plant Physiology 2016, 171, 566-579, doi:10.1104/pp.16.00127.
- Pantoja, O. High affinity ammonium transporters: molecular mechanism of action. Front Plant Sci 2012, 3, 34, doi:10.3389/fpls.2012.00034.
- Schmitz, A.M.; Harrison, M.J. Signaling events during initiation of arbuscular mycorrhizal symbiosis. Journal of Integrative Plant Biology 2014, 56, 250-261, doi:doi:10.1111/jipb.12155.
- Streng, A.; op den Camp, R.; Bisseling, T.; Geurts, R. Evolutionary origin of Rhizobium Nod factor signaling. Plant Signaling & Behavior 2011, 6, 1510-1514, doi:10.4161/psb.6.10.17444.
- Limpens, E.; Ramos, J.; Franken, C.; Raz, V.; Compaan, B.; Franssen, H.; Bisseling, T.; Geurts, R. RNA interference in Agrobacterium rhizogenes-transformed roots of Arabidopsis and Medicago truncatula. J. Exp. Bot. 2004, 55, 983-992, doi:10.1093/jxb/erh122.
- Deguchi, Y.; Banba, M.; Shimoda, Y.; Chechetka, S.A.; Suzuri, R.; Okusako, Y.; Ooki, Y.; Toyokura, K.; Suzuki, A.; Uchiumi, T.; et al. Transcriptome profiling of Lotus japonicus roots during arbuscular mycorrhiza development and comparison with that of nodulation. DNA Res 2007, 14, 117-133, doi:10.1093/dnares/dsm014.

Reviewer 2 Report
The Manuscript ijms-2539762 by Ovchinnikova et al. is dealing with the regulation of arbuscular mycorrhizal symbiosis in Medicago truncatula. Here, the Authors describe the role of a transcription factor involved in expression of a potential ammonium transporter from the plant, but also of several further transporters. The topic of the study is interesting as nutrient transport plays an important role in symbiotic exchange and in the functioning of the mycorrhizal association.
Major points:
(1) From the presented data, specificity of the reported transcription factor becomes not really clear. So, finally, even that binding to an AMF promoter was shown in Fig. 6E, the question remains, which effects are direct or indirect?
(2) From the beginning, AMF is reported as an ammonium transporter ("facilitator"), but functional evidence is much less reported than for the AMT family. Already in the Introduction and also later (eg lines 156-157, also within the Discussion), the AMT family and the AMF are discussed somehow together, without clearly distinguishing and describing that these are two different families. Redundance and/or specificity of this function mediated potentially by two different families with several members should be discussed.
(3) Further, functional evidence is reported by an indirect yeast assay (toxicity and growth inhibition, Fig. 6A). However, in this figure growth inhibition seems to be rather similar in control conditions than when adding methylammonium? Already in control conditions, the figure shows reduced growth, but this is not commented? Moreover, the Authors state, that MtAMF1;3 would transport "specifically" MA (line 305), but this has never been shown and is in contrast to statements within the Introduction ("DHA2 family of drug:H+ antiporters", line 116).
(4) The link to Pi, mentioned first in the Introduction in lines129-130, ist not really clear. Moreover, the reported data seem somehow to be contradictory between Pi-dependence and induction by AM symbiosis? Description in lines 190-196 is not really clear, the result describes non-myco or myco roots? Further, if the TF is normally induced by AM symbiosis but here more significantly by high Pi what would suppress mycorhization, it seems that these are contrasting results??
(5) "The AM enhanced growth was significantly reduced with RNAi silencing... " (line 222), but Fig 2B shows unchanged or even improved colonization at high Pi?? This is contradictory...? Moreover, figures 3C are in contrast to 2B?
(6) "... were found actvated in both non-mycorrhial and mycorrhizal roots" (line 245), this is in contrast to earlier statements that theses are symbiosis-induced genes?
(7) Transport direction of ammonium from the Conclusion and model in Suppl fig. 3) is not really clear?
Minor points:
(8) At which membrane is the AMF expressed?? Is it really plasma membrane? ("intracellular NH4+ homeostasis" has been mentioned for yeast in line 368?)
(9) Two and three members are reported in lines 144-147. What is the reason for the selection of one AMF and one TF?
(10) Mycorrhization is not given (line 154)??
(11) From the expression data (Figure 1), given as fold changes, it becomes not really what might be the expression level of these candidates already in the non-mycorrhizal plant?
(12) Overall, the Manuscript is well written, however, some precisions and little corrections are needed, e.g. phrases lines 66, 145, 384-387; et al. (twice, line 66), space (lines 69, 108), ... delete "channel" (line 359), this transporter is not a channel!
(13) Species names should be always written in italic (e.g. lines 141, 153, 288), first time full, later abbreviated; Medicago (given in line 20) has changed later in the Discussion as Medicago.
(14) Protein & gene names should be unified, MtXX and MtXX, and not changed within the mansucript (MT, Mt)...
(15) Finally, maybe, from the beginning, it should be more clearly stated, what was already known, and the Conclusion could better summarize the new progress.
Overall, the Manuscript is well written, however, some precisions and little corrections are needed, e.g. phrases lines 66, 145, 384-387; et al. (twice, line 66), space (lines 69, 108), ... delete "channel" (line 359), this transporter is not a channel!
Species names should be always written in italic (e.g. lines 141, 153, 288), first time full, later abbreviated; Medicago (given in line 20) has changed later in the Discussion as Medicago.
Protein & gene names should be unified, MtXX and MtXX, and not changed within the mansucript (MT, Mt)...
Author Response
Response to Reviewer Comments
Reviewer 2
The Manuscript ijms-2539762 by Ovchinnikova et al. is dealing with the regulation of arbuscular mycorrhizal symbiosis in Medicago truncatula. Here, the Authors describe the role of a transcription factor involved in expression of a potential ammonium transporter from the plant, but also of several further transporters. The topic of the study is interesting as nutrient transport plays an important role in symbiotic exchange and in the functioning of the mycorrhizal association.
Major points:
(1) From the presented data, specificity of the reported transcription factor becomes not really clear. So, finally, even that binding to an AMF promoter was shown in Fig. 6E, the question remains, which effects are direct or indirect?
Response 1: At this stage of the investigation, we do not know whether there is a direct link to the promoter of MtAMF1;3 by MtbHLHm1;1. We can show an affinity of the TF to the promoter of MtAMF1;3 in vitro but have yet to confirm this in planta. However, the correlative responses between MtbHLHm1 and Mt AMF1;3 would suggest a possible interaction.
(2) From the beginning, AMF is reported as an ammonium transporter ("facilitator"), but functional evidence is much less reported than for the AMT family. Already in the Introduction and also later (eg lines 156-157, also within the Discussion), the AMT family and the AMF are discussed somehow together, without clearly distinguishing and describing that these are two different families. Redundance and/or specificity of this function mediated potentially by two different families with several members should be discussed.
Response 2: We describe the two different ammonium transport families in the introduction (lines 82-139). We have introduced a extension of the sentence in line 67 to highlight the AMT/MEP/Rhesus transport family.
“a member of the high-affinity (AMT/MEP/Rhesus) transport family [13].”
AMF proteins are described at lines 121-132
(3) Further, functional evidence is reported by an indirect yeast assay (toxicity and growth inhibition, Fig. 6A). However, in this figure growth inhibition seems to be rather similar in control conditions than when adding methylammonium? Already in control conditions, the figure shows reduced growth, but this is not commented? Moreover, the Authors state, that MtAMF1;3 would transport "specifically" MA (line 305), but this has never been shown and is in contrast to statements within the Introduction ("DHA2 family of drug:H+ antiporters", line 116).
Response 3: The MA toxicity shown on plates (Figure 6A) highlights a reduction in growth of the three AMF lines as cell concentrations are serial diluted. The growth is accumulative on plates but there is a clear difference in the level of growth from the pYES 3 control as the cells are diluted. This data is supported by liquid growth cultures which show a clear reduction in cell division with MA in the media for cells expressing ScAMF1, MtAMF1;3 and MtbHLHm1. Significant increases in net 14C-MA uptake into yeast cells confirms MA is transported into the yeast cells when expressing either ScAMF1, MtAMF1;3 and MtbHLHm1. These results are typical for AMF activities in yeast cells – see Chiasson et al [6].
We tested whether MtAMF1;3 could transport MA as an analogue of actual NH4+ transport. We have previously shown that GmAMF1;3 and ScAMF1 transport MA in yeast cells and Xenopus laevis oocytes [6], using multiple approaches. AMF proteins are homologs of the DHA2 family of drug:H+ antiporters, this has no bearing on whether they transport MA or not. As highlighted in the manuscript, the substrate these transporters facilitate is increasingly broad.
(4) The link to Pi, mentioned first in the Introduction in lines129-130, ist not really clear. Moreover, the reported data seem somehow to be contradictory between Pi-dependence and induction by AM symbiosis? Description in lines 190-196 is not really clear, the result describes non-myco or myco roots? Further, if the TF is normally induced by AM symbiosis but here more significantly by high Pi what would suppress mycorhization, it seems that these are contrasting results??
Response 4: The regulatory controls between the Rhizobial and AM fungal symbioses are increasingly being found to be similar through shared TF signalling cascades and the presence of like for like proteins controlling invasion [14,15]. We believe the similarity is a worthy reason to explore relationships with previously identified gene networks operating in the legume rhizobial sysmbiosis.
The relationship between Pi induction of MtbHLHm1;1 expression and AM colonisation is interesting. It’s quite clear the expression of MtbHLHm1 is important for the symbiosis to function but its activation with Pi may suggest a different signalling pathway that elevated Pi is activating on contrary to the negative impact it generally has on setting up a AM symbiosis. The interaction may be related to the ammonium transporter proteins which appear to be positively regulated by MtbHLHm1;1 expression in response to AM colonisation.
(5) "The AM enhanced growth was significantly reduced with RNAi silencing... " (line 222), but Fig 2B shows unchanged or even improved colonization at high Pi?? This is contradictory...? Moreover, figures 3C are in contrast to 2B?
Response 5: We have addressed these discrepancies and revised Figures 2 and 3 – See Reviewer 1, response 12.
(6) "... were found actvated in both non-mycorrhial and mycorrhizal roots" (line 245), this is in contrast to earlier statements that theses are symbiosis-induced genes?
Response 6: This context refers to their response in gene expression after AM inoculation. The text has been changed from activated to active.
(7) Transport direction of ammonium from the Conclusion and model in Suppl fig. 3) is not really clear?
Response 7: We don’t know the transport direction of ammonium by AMF proteins in planta. We expect it involves efflux across the periarbuscular membrane to the plant, but it is too early to make this conclusion. We have highlighted in the legend of Supplementary Figure 4 that direction of transport has yet to be defined.
Minor points:
(8) At which membrane is the AMF expressed?? Is it really plasma membrane? ("intracellular NH4+ homeostasis" has been mentioned for yeast in line 368?)
Response 8: We don’t have an exhaustive overview of membrane localisation in this study. Our GFP tagged MtAMF1;3 suggests a location at the plasma membrane. We have previously reported plasma membrane localisation for GmAMF1;3 and ScAMF1.
(9) Two and three members are reported in lines 144-147. What is the reason for the selection of one AMF and one TF?
Response 9: We chose to follow the activities of MtbHLhm1;1 and MtAMF1;3 as a resource and workload decision. Follow on experiments will reveal the roles of other AMF and bHLHm1 homologs.
(10) Mycorrhization is not given (line 154)??
Response 10: Plants were provided mycorrhizal spores from Glomus etunicatum, Glomus coronatum, Glomus intraradices and Glomus mosseae. We hope this is the question being raised in question 10?
(11) From the expression data (Figure 1), given as fold changes, it becomes not really what might be the expression level of these candidates already in the non-mycorrhizal plant?
Response 11: Our intention in using fold-changes in gene expression was to explore how + AM or ± MtbHLHm1;1 influences other gene activities. This type of data presentation is common and widely accepted to highlight changes in gene expression.
(12) Overall, the Manuscript is well written, however, some precisions and little corrections are needed, e.g. phrases lines 66, 145, 384-387; et al. (twice, line 66), space (lines 69, 108), ... delete "channel" (line 359), this transporter is not a channel!
Response 12: We have evaluated the text and made numerous corrections as identified by reviewers and the authors. We will delete the word channel as it still remains unclear how these transport proteins function.
(13) Species names should be always written in italic (e.g. lines 141, 153, 288), first time full, later abbreviated; Medicago (given in line 20) has changed later in the Discussion as Medicago.
(14) Protein & gene names should be unified, MtXX and MtXX, and not changed within the mansucript (MT, Mt)...
Response 13 & 14: Corrected throughout the manuscript.
(15) Finally, maybe, from the beginning, it should be more clearly stated, what was already known, and the Conclusion could better summarize the new progress.
Response 15: We have made significant edits to this manuscript covering its introduction, interpretation of the results and how the data is discussed.
References:
- Breuillin-Sessoms, F.; Floss, D.S.; Gomez, S.K.; Pumplin, N.; Ding, Y.; Levesque-Tremblay, V.; Noar, R.D.; Daniels, D.A.; Bravo, A.; Eaglesham, J.B.; et al. Suppression of arbuscule degeneration in Medicago truncatula phosphate transporter4 mutants is dependent on the ammonium transporter 2 family protein AMT2;3. The Plant Cell 2015, 27, 1352-1366, doi:10.1105/tpc.114.131144.
- Vierheilig, H.; Coughlan, A.P.; Wyss, U.; Piche, Y. Ink and vinegar, a simple staining technique for arbuscular-mycorrhizal fungi. Appl. Environ. Microbiol. 1998, 64, 5004-5007.
- McGonigle, T.P.; Miller, M., H.; Evans, D., G.; Fairchild, G., L.; Swan, J., A. A new method which gives an objective measure of colonization of roots by vesicular—arbuscular mycorrhizal fungi. New Phytol. 1990, 115, 495-501, doi:10.1111/j.1469-8137.1990.tb00476.x.
- Trouvelot, A. Mesure du taux de mycorhization VA d'un systeme radiculaire. Recherche de methodes d'estimation ayant une significantion fonctionnelle. Mycorrhizae : physiology and genetics 1986, 217-221.
- Harrison, M.J.; Dewbre, G.R.; Liu, J. A phosphate transporter from Medicago truncatula involved in the acquisition of phosphate released by arbuscular mycorrhizal fungi. The Plant Cell 2002, 14, 2413-2429.
- Chiasson, D.M.; Loughlin, P.C.; Mazurkiewicz, D.; Mohammadidehcheshmeh, M.; Fedorova, E.E.; Okamoto, M.; McLean, E.; Glass, A.D.M.; Smith, S.E.; Bisseling, T.; et al. Soybean SAT1 (Symbiotic Ammonium Transporter 1) encodes a bHLH transcription factor involved in nodule growth and NH4+ transport. Proceedings of the National Academy of Sciences 2014, 111, 4814-4819, doi:10.1073/pnas.1312801111.
- Guether, M.; Neuhäuser, B.; Balestrini, R.; Dynowski, M.; Ludewig, U.; Bonfante, P. A mycorrhizal-specific ammonium transporter from Lotus japonicus acquires nitrogen released by arbuscular mycorrhizal fungi. Plant Physiol. 2009, 150, 73-83, doi:10.1104/pp.109.136390.
- Deng, J.; Zhu, F.G.; Liu, J.X.; Zhao, Y.F.; Wen, J.Q.; Wang, T.; Dong, J.L. Transcription Factor bHLH2 Represses CYSTEINE PROTEASE77 to Negatively Regulate Nodule Senescence. Plant Physiology 2019, 181, 1683-1703, doi:10.1104/pp.19.00574.
- Breuillin, F.; Schramm, J.; Hajirezaei, M.; Ahkami, A.; Favre, P.; Druege, U.; Hause, B.; Bucher, M.; Kretzschmar, T.; Bossolini, E.; et al. Phosphate systemically inhibits development of arbuscular mycorrhiza in Petunia hybrida and represses genes involved in mycorrhizal functioning. The Plant Journal 2010, 64, 1002-1017, doi:10.1111/j.1365-313X.2010.04385.x.
- Balzergue, C.; Chabaud, M.; Barker, D.G.; Becard, G.; Rochange, S.F. High phosphate reduces host ability to develop arbuscular mycorrhizal symbiosis without affecting root calcium spiking responses to the fungus. Frontiers in plant science 2013, 4, 426, doi:10.3389/fpls.2013.00426.
- Nouri, E.; Breuillin-Sessoms, F.; Feller, U.; Reinhardt, D. Phosphorus and nitrogen regulate arbuscular mycorrhizal symbiosis in Petunia hybrida. PLOS ONE 2014, 9, e90841, doi:10.1371/journal.pone.0090841.
- Kobae, Y.; Ohmori, Y.; Saito, C.; Yano, K.; Ohtomo, R.; Fujiwara, T. Phosphate treatment strongly inhibits new arbuscule development but not the maintenance of arbuscule in mycorrhizal rice roots. Plant Physiology 2016, 171, 566-579, doi:10.1104/pp.16.00127.
- Pantoja, O. High affinity ammonium transporters: molecular mechanism of action. Front Plant Sci 2012, 3, 34, doi:10.3389/fpls.2012.00034.
- Schmitz, A.M.; Harrison, M.J. Signaling events during initiation of arbuscular mycorrhizal symbiosis. Journal of Integrative Plant Biology 2014, 56, 250-261, doi:doi:10.1111/jipb.12155.
- Streng, A.; op den Camp, R.; Bisseling, T.; Geurts, R. Evolutionary origin of Rhizobium Nod factor signaling. Plant Signaling & Behavior 2011, 6, 1510-1514, doi:10.4161/psb.6.10.17444.
- Limpens, E.; Ramos, J.; Franken, C.; Raz, V.; Compaan, B.; Franssen, H.; Bisseling, T.; Geurts, R. RNA interference in Agrobacterium rhizogenes-transformed roots of Arabidopsis and Medicago truncatula. J. Exp. Bot. 2004, 55, 983-992, doi:10.1093/jxb/erh122.
- Deguchi, Y.; Banba, M.; Shimoda, Y.; Chechetka, S.A.; Suzuri, R.; Okusako, Y.; Ooki, Y.; Toyokura, K.; Suzuki, A.; Uchiumi, T.; et al. Transcriptome profiling of Lotus japonicus roots during arbuscular mycorrhiza development and comparison with that of nodulation. DNA Res 2007, 14, 117-133, doi:10.1093/dnares/dsm014.
Reviewer 3 Report
This is a comprehensive work about the role transcription Factor MTBHLHM1 and the ammonium facilitator protein MTAMF1;3 during arbuscular-mycorrhizal symbiosis in Medicago truncatula.
However I have some questions and comments:
Line 62 and everywhere else. Form “Xuan et al. (19)” is commonly used as a reference. This is not good to use this way of mentioning researcher’s name in the text. It is better to say “Xuan with authors”. In fact, there are numerous references in the text, so there is no need frequently to mention researcher’s names.
Line 93-96 – in my opinion these methodological details are not necessary in the introduction.
According lines 154-162, figure 1A deals with non-inoculated and inoculated wild-type roots. However the caption of figure 1A says that the picture shows the expression in empty vector roots. So what does figure 1A show?
Uniform this, either you use ‘non-inoculated’ or non-mycorrhizal’ or ‘non-colonized’ because the meaning might be different. Check this everywhere in the text and captions for figures 4, 5.
Lines 195 and 199 – what is “WT roots” and what is “non-mycorrhizal roots”.
What is mycorrhizal growth response? How did you evaluate mycorrhizal colonization, mycorrhizal growth response and degraded arbuscules?
Check the figure S4. In the text ‘non-inoculated’ is used, but in the caption and on the picture the word ‘uninoculated’ is used. Uniform this. Also, there are only 3 supplementary figures.
Check the caption for the table 1.
Microscopy pictures are very so so…figures 4 A, B, 5 A are overexposed. In figure 5 B brightness/contrast are not adjusted.
Lines 246-247 – “GUS expression for both MtbHLHm1 and MtAMF1;3 was diffuse across the epidermis, root hair, cortex and stele (Figures 4A and 5A, respectively)” it means everywhere. How did you check if it is specific to MtbHLHm1 and MtAMF1;3? Root hair is not clear on figure 4A, B.
Lines 248-250 – “However, in mycorrhizal colonized roots, GUS
staining was more specific and localised to the stele and in the cortex (Figures 4B, 4C) where AM fungi were also identified using secondary ink-staining (Figures 4C and 5–E).” on figure 5E, I see root cells stained with WGA Texas red. To combine two blue colors on one picture is not the best choice. I do not see GUS staining on ink-stained roots.
Lines 251-259 – check the references for figures.
Figures 4 and 5. If several colors are used. Use should say what color is what.
Fig 4G is GFP, and what about D-F? what is grey color? On fig 4 G—I, I see black spot instead of arbuscule. How do you know that this black spot on a picture is arbuscule? The same situation is with root cortex on fig 5 K-M. How do you distinguish root cortex cells on this black area?
Fig 4F contains ‘v’ but there is no definition of v is the caption.
Materials and methods:
What substrate was used to grow plants?
Give the reference or the whole protocols of preparing sections using Technovit.
Give the manufacturer for a rotary microtome.
Give the protocol of staining with WGA. WGA Texas red is produced by Thermo fisher. Did you use anti-WGA antibodies or WGA Texas red?
Check line 696 – who is XX and YY?
Author Response
This is a comprehensive work about the role transcription Factor MTBHLHM1 and the ammonium facilitator protein MTAMF1;3 during arbuscular-mycorrhizal symbiosis in Medicago truncatula.
However I have some questions and comments:
1) Line 62 and everywhere else. Form “Xuan et al. (19)” is commonly used as a reference. This is not good to use this way of mentioning researcher’s name in the text. It is better to say “Xuan with authors”. In fact, there are numerous references in the text, so there is no need frequently to mention researcher’s names.
Response 1: The use of a first author is common in scientific writing and provides necessary acknowledgement to the authors. The et al., refers to the other authors.
2) Line 93-96 – in my opinion these methodological details are not necessary in the introduction.
Response 2: We have included some technical detail in the introduction to highlight the type of experiment that was used to generate this specific result. This is a similar process used in these experiments and therefore provides a useful reference and context for the paper, we would like to retain this text as is.
3) According lines 154-162, figure 1A deals with non-inoculated and inoculated wild-type roots. However the caption of figure 1A says that the picture shows the expression in empty vector roots. So what does figure 1A show?
Response 3: This is an misinterpretation. The control plants were are all transformed with an empty vector and used as the control. We have modified the text, see below.
“Transgenic (hairy-root) plant roots were evaluated 8 weeks after inoculation (WAI). In empty vector (control) roots, MtbHLHm1;1, MtAMF1;3 and MtAMF2;1 expression were significantly (P<0.05) enhanced when inoculated with AM fungi (Figure 1A).”
4) Uniform this, either you use ‘non-inoculated’ or non-mycorrhizal’ or ‘non-colonized’ because the meaning might be different. Check this everywhere in the text and captions for figures 4, 5.
Response 4: Thanks for highlighting these terminology differences. We have converted all text to be non-inoculated when referred to as -AM.
5) Lines 195 and 199 – what is “WT roots” and what is “non-mycorrhizal roots”.
Response 5: This has been addressed in Response 3.
6) What is mycorrhizal growth response? How did you evaluate mycorrhizal colonization, mycorrhizal growth response and degraded arbuscules?
Response 6: We have addressed this in a similar question raised by reviewer 1. Please seeResponse 11 (above)
“Non-inoculated and inoculated Medicago plants were dried overnight at 60°C and weighed to measure total plant biomass. %MGR was calculated based on the differences in DW between non-inoculated and inoculated Medicago plants.”
7) Check the figure S4. In the text ‘non-inoculated’ is used, but in the caption and on the picture the word ‘uninoculated’ is used. Uniform this. Also, there are only 3 supplementary figures.
Response 7: Corrected.
8) Check the caption for the table 1.
Response 8: An expanded legend has been added to Supplementary Table 1.
9) Microscopy pictures are very so so…figures 4 A, B, 5 A are overexposed. In figure 5 B brightness/contrast are not adjusted.
10) Lines 246-247 – “GUS expression for both MtbHLHm1 and MtAMF1;3 was diffuse across the epidermis, root hair, cortex and stele (Figures 4A and 5A, respectively)” it means everywhere. How did you check if it is specific to MtbHLHm1 and MtAMF1;3? Root hair is not clear on figure 4A, B.
Response 9 and 10: Unfortunately, the images chosen were the best obtained at the time of experimentation. There is no opportunity in the project to go back and re-create these plant lines to obtain higher quality images. In Figure 4A and 5A, we created gene specific promoter fusions with GUS. We assume this expression is the representative expression pattern of these two genes using GUS as an assay. From previous work we know both GmbHLHm1 and GmAMF3 are expressed in cells aligned with the stele in non-inoculated roots (see Chiasson et al [6]).
11) Lines 248-250 – “However, in mycorrhizal colonized roots, GUS
staining was more specific and localised to the stele and in the cortex (Figures 4B, 4C) where AM fungi were also identified using secondary ink-staining (Figures 4C and 5–E).” on figure 5E, I see root cells stained with WGA Texas red. To combine two blue colors on one picture is not the best choice. I do not see GUS staining on ink-stained roots.
Response 11: In Figure 4 A-C, GUS expression for MtbHLHm1 is seen localized to arbuscule containing cells in the cortex approaching the stele. The use of an Indian-ink counter stain shows the presence of arbuscules in cortical cells (Figure 4C). We agree, the combined blue colored stains hides detail, but was necessary to define the arbuscule structures in the GUS stained background. In subsequent experiments with MtAMF1;3, we used both GUS (Figure 5 A-C) and a WGA-Texas Red antibody to observe arbuscules (Figure 5 D&E). We have modified the text in the manuscript to correct this error (see below).
“Non-inoculated plants of similar age were used as a control. MtbHLHm1;1pro and MtAMF1;3pro were found active in both non-inoculated and mycorrhizal roots (Figures 4A, B and 5A, B, respectively). In the non-inoculated control roots, GUS expression for both MtbHLHm1;1 and MtAMF1;3 was diffuse across the epidermis, root hair, cortex and stele (Figures 4A and 5A, respectively). In mycorrhizal colonised roots, GUS staining was more specific and localised in cortical cells and the stele (Figure 4B), where AM fungi were also identified using secondary ink-staining (Figures 4C) or with WGA-Texas Red antibody (Figure 5 D,E).”
12) Lines 251-259 – check the references for figures.
13) Figures 4 and 5. If several colors are used. Use should say what color is what.
14) Fig 4G is GFP, and what about D-F? what is grey color? On fig 4 G—I, I see black spot instead of arbuscule. How do you know that this black spot on a picture is arbuscule? The same situation is with root cortex on fig 5 K-M. How do you distinguish root cortex cells on this black area?
15) Fig 4F contains ‘v’ but there is no definition of v is the caption.
Response 12-15: We have addressed each of these concerns identified in questions 12-15. We corrected the Figure numbers (12) in the identified text. We are not sure what is meant by colour or specifying what colour as suggested in question 13. We have identified ‘v’ as vcuole in the legend of Figure 4. The grey colour in Figure 5 F and 5H represents a DIC image (Differential Interference Contrast). The black spot that is referred to represents the general arbuscule containing cell. We have moved the Arb label to sit closer to the branched hyphae of the arbuscule for both Figures 4 and 5.
Materials and methods:
What substrate was used to grow plants?
Response: For plate based assays, we grew seedlings on Fahraeus medium. Plants grown in soil or in liquid culture were supplied a liquid growth media using recipes defined by Limpens et al [16].
Give the reference or the whole protocols of preparing sections using Technovit.
Response: we have included the protocol used as described by Deguchi et al., 2007 [17].
Give the manufacturer for a rotary microtome.
Response: This was included in the manuscript – Leica RM2255..
Give the protocol of staining with WGA. WGA Texas red is produced by Thermo fisher. Did you use anti-WGA antibodies or WGA Texas red?
Response: The protocol is highlighted in lines 614-616 and includes a methods reference Harrison et al [5]. We used Texas-Red sourced from Sigma-Aldrich. We used anti-WGA antibodies linked to the TexasRed fluorophore.
Check line 696 – who is XX and YY?
Response: E.O. and B.N.K.
References:
- Breuillin-Sessoms, F.; Floss, D.S.; Gomez, S.K.; Pumplin, N.; Ding, Y.; Levesque-Tremblay, V.; Noar, R.D.; Daniels, D.A.; Bravo, A.; Eaglesham, J.B.; et al. Suppression of arbuscule degeneration in Medicago truncatula phosphate transporter4 mutants is dependent on the ammonium transporter 2 family protein AMT2;3. The Plant Cell 2015, 27, 1352-1366, doi:10.1105/tpc.114.131144.
- Vierheilig, H.; Coughlan, A.P.; Wyss, U.; Piche, Y. Ink and vinegar, a simple staining technique for arbuscular-mycorrhizal fungi. Appl. Environ. Microbiol. 1998, 64, 5004-5007.
- McGonigle, T.P.; Miller, M., H.; Evans, D., G.; Fairchild, G., L.; Swan, J., A. A new method which gives an objective measure of colonization of roots by vesicular—arbuscular mycorrhizal fungi. New Phytol. 1990, 115, 495-501, doi:10.1111/j.1469-8137.1990.tb00476.x.
- Trouvelot, A. Mesure du taux de mycorhization VA d'un systeme radiculaire. Recherche de methodes d'estimation ayant une significantion fonctionnelle. Mycorrhizae : physiology and genetics 1986, 217-221.
- Harrison, M.J.; Dewbre, G.R.; Liu, J. A phosphate transporter from Medicago truncatula involved in the acquisition of phosphate released by arbuscular mycorrhizal fungi. The Plant Cell 2002, 14, 2413-2429.
- Chiasson, D.M.; Loughlin, P.C.; Mazurkiewicz, D.; Mohammadidehcheshmeh, M.; Fedorova, E.E.; Okamoto, M.; McLean, E.; Glass, A.D.M.; Smith, S.E.; Bisseling, T.; et al. Soybean SAT1 (Symbiotic Ammonium Transporter 1) encodes a bHLH transcription factor involved in nodule growth and NH4+ transport. Proceedings of the National Academy of Sciences 2014, 111, 4814-4819, doi:10.1073/pnas.1312801111.
- Guether, M.; Neuhäuser, B.; Balestrini, R.; Dynowski, M.; Ludewig, U.; Bonfante, P. A mycorrhizal-specific ammonium transporter from Lotus japonicus acquires nitrogen released by arbuscular mycorrhizal fungi. Plant Physiol. 2009, 150, 73-83, doi:10.1104/pp.109.136390.
- Deng, J.; Zhu, F.G.; Liu, J.X.; Zhao, Y.F.; Wen, J.Q.; Wang, T.; Dong, J.L. Transcription Factor bHLH2 Represses CYSTEINE PROTEASE77 to Negatively Regulate Nodule Senescence. Plant Physiology 2019, 181, 1683-1703, doi:10.1104/pp.19.00574.
- Breuillin, F.; Schramm, J.; Hajirezaei, M.; Ahkami, A.; Favre, P.; Druege, U.; Hause, B.; Bucher, M.; Kretzschmar, T.; Bossolini, E.; et al. Phosphate systemically inhibits development of arbuscular mycorrhiza in Petunia hybrida and represses genes involved in mycorrhizal functioning. The Plant Journal 2010, 64, 1002-1017, doi:10.1111/j.1365-313X.2010.04385.x.
- Balzergue, C.; Chabaud, M.; Barker, D.G.; Becard, G.; Rochange, S.F. High phosphate reduces host ability to develop arbuscular mycorrhizal symbiosis without affecting root calcium spiking responses to the fungus. Frontiers in plant science 2013, 4, 426, doi:10.3389/fpls.2013.00426.
- Nouri, E.; Breuillin-Sessoms, F.; Feller, U.; Reinhardt, D. Phosphorus and nitrogen regulate arbuscular mycorrhizal symbiosis in Petunia hybrida. PLOS ONE 2014, 9, e90841, doi:10.1371/journal.pone.0090841.
- Kobae, Y.; Ohmori, Y.; Saito, C.; Yano, K.; Ohtomo, R.; Fujiwara, T. Phosphate treatment strongly inhibits new arbuscule development but not the maintenance of arbuscule in mycorrhizal rice roots. Plant Physiology 2016, 171, 566-579, doi:10.1104/pp.16.00127.
- Pantoja, O. High affinity ammonium transporters: molecular mechanism of action. Front Plant Sci 2012, 3, 34, doi:10.3389/fpls.2012.00034.
- Schmitz, A.M.; Harrison, M.J. Signaling events during initiation of arbuscular mycorrhizal symbiosis. Journal of Integrative Plant Biology 2014, 56, 250-261, doi:doi:10.1111/jipb.12155.
- Streng, A.; op den Camp, R.; Bisseling, T.; Geurts, R. Evolutionary origin of Rhizobium Nod factor signaling. Plant Signaling & Behavior 2011, 6, 1510-1514, doi:10.4161/psb.6.10.17444.
- Limpens, E.; Ramos, J.; Franken, C.; Raz, V.; Compaan, B.; Franssen, H.; Bisseling, T.; Geurts, R. RNA interference in Agrobacterium rhizogenes-transformed roots of Arabidopsis and Medicago truncatula. J. Exp. Bot. 2004, 55, 983-992, doi:10.1093/jxb/erh122.
- Deguchi, Y.; Banba, M.; Shimoda, Y.; Chechetka, S.A.; Suzuri, R.; Okusako, Y.; Ooki, Y.; Toyokura, K.; Suzuki, A.; Uchiumi, T.; et al. Transcriptome profiling of Lotus japonicus roots during arbuscular mycorrhiza development and comparison with that of nodulation. DNA Res 2007, 14, 117-133, doi:10.1093/dnares/dsm014.
Round 2
Reviewer 1 Report
Few points remain for the revised manuscript.
The induction of especially MtAMF1;3 is not obvious from the presented promoter-GUS images. Also for bHLHm1 more detailed images could be provided. Unfortunately, the authors did not make use of available RNAseq data and gene chip data to support their claims.
Similarly, the membrane association of the proteins is not at all obvious from the presented images. I would propose to tone down the formulations on the membrane localizations, or otherwise provide more clear images and controls for the membrane association. At best I would say that it can only be concluded that both bHLHm1 and AMF1;3 are both expressed in arbuscule containing cells. But they may also have functions in other cell types.
Unfortunately, the effect of AMF1;3 silencing on arbuscule morphology (or % degrading arbuscules) is not shown.
Author Response
The induction of especially MtAMF1;3 is not obvious from the presented promoter-GUS images. Also for bHLHm1 more detailed images could be provided. Unfortunately, the authors did not make use of available RNAseq data and gene chip data to support their claims.
Response: We have used and presented qPCR data to provide quantitative expression levels of MtAMF1;3 and MtbHLHm1 in response to AM colonisation (Figure 1A). In this experiment, +AM enhances both genes relative to the non-inoculated plants. The promoter::GUS images (Figures 4 and 5) were designed to show tissue localisation rather than quantitative levels of expression. We believe it is difficult to capture linearity in GUS expression equally across all cell types in multicellular tissues. We haven't included public data sets showing RNA-seq experiments in Medicago in response to AM colonisation. We feel the experimental data is rigorous enough to stand on its own and furthermore reflects the experimental systems we were using in this study.
"Quantitative expression levels of GUS was not measured in these plants, instead we used more sensitive mRNA transcript abundance (qPCR) assays to record the positive influences on MtbHLHm1 and MtAMF1;3 gene expression by AM colonisation (Figure 1)."
Similarly, the membrane association of the proteins is not at all obvious from the presented images. I would propose to tone down the formulations on the membrane localizations, or otherwise provide more clear images and controls for the membrane association. At best I would say that it can only be concluded that both bHLHm1 and AMF1;3 are both expressed in arbuscule containing cells. But they may also have functions in other cell types.
Response: We have identified possible membrane locations when using GFP tagged proteins. We agree, our use of definitive labels should be presented in a more conservative tone in the manuscript, since this study didn't confirm intercellular localisation using purified protein preps and protein-specific antibodies. We haven't said these are specific arbuscule proteins and believe they have functions elsewhere in the cells. We have made changes in the text to reflect this yet to be complete analysis and interpretation of membrane localisation.
"The the lack of protein-specific antibodies for this study, has limited our ability to conclusively identify the intercellular membrane locations of both MtbHLHm1 and MtAMF1;3." - lines 291-293.
Unfortunately, the effect of AMF1;3 silencing on arbuscule morphology (or % degrading arbuscules) is not shown.
Response: Unfortunately, when this experiment was completed the % degraded arbuscules was not counted for the Mtamf1;3 lines.
Reviewer 3 Report
The text is improved.
Except on part. Harrison et al 2002 used WGA –Texas red from Molecular probes (now Thermo fisher). Harrison used this dye to counterstain fungi during immunolocalization of MtPT4. If anti-WGA antibodies were used, give a technique. Besides in Sigma I found only unconjugated anti-WGA antibodies. However if you used antibodies but the technique of immunolocalization was taken from Harrison, it should be mention in the text.
About colors on pictures. Usually if figures contain pictures with different colors, it is mention in a caption as channels. For example red channel – fungi stained with Texas red. Green channel – localization of GFP label. Because all the pictures obtained with confocal microscopy can be shown with different colors. GFP might be blue and wga Texas red might be magenta.
Author Response
Except on part. Harrison et al 2002 used WGA –Texas red from Molecular probes (now Thermo fisher). Harrison used this dye to counterstain fungi during immunolocalization of MtPT4. If anti-WGA antibodies were used, give a technique. Besides in Sigma I found only unconjugated anti-WGA antibodies. However if you used antibodies but the technique of immunolocalization was taken from Harrison, it should be mention in the text.
Response: We have edited the methods section to highlight procedures used with the anti-WGA antibodies. We have also removed the Harrison et al 2002 reference. Changes can be found at lines 633-642.
"Roots were fixed in 1% (v/v) of freshly prepared paraformaldehyde in 1x PBS (v/v), pH 7.4, for 30 min at 4°C, then washed 3 times in 1x PBS and embedded in 3% (w/v) low melting point Agaros (Sigma-Aldrich) in 1x PBS. Embedded roots were sectioned with a vibrotome VT1200S (Leica Biosystems, Nussloch, Germany) at 100 µm thick and blocked in 3% BSA (20 min room temperature) with following washing in 1x PBS. To label fungi inside roots and inside arbuscule-containing cells, 100 µm Agarose root sections were incubated with Wheat Germ Agglutinin (WGA) antibody linked to a TexasRed® fluorophore (Sigma-Aldrich), prepared and used according to the supplier’s instructions. WGA-TexasRed® labelled root sections were mounted on glass slides and analyses by confocal microscopy."
About colors on pictures. Usually if figures contain pictures with different colors, it is mention in a caption as channels. For example red channel – fungi stained with Texas red. Green channel – localization of GFP label. Because all the pictures obtained with confocal microscopy can be shown with different colors. GFP might be blue and wga Texas red might be magenta.
Response: Apologies, we have now redefined the descriptions of the colours presented in Figures 4 and 5. We have updated the two legends to identify the colours presented. The excitation wavelengths to identify either GFP and DS-Red are described in the materials and methods - Lines 642-648.